# How Do Language Models Speak Languages?
# A Case Study on Unintended Code-Switching

**Yuxin Xiao**[* 1]  **Zhen Huang**[2]  **Wenxiao Wang**[3]  **Yan Zhao**[4]  **Zhihong Gu**[5]  **Binbin Lin**[3]
**Xiaofei He**[1 6]  **Xu Shen**[2]  **Jieping Ye**[2]

## Abstract

Unintended code-switching, where LLMs unexpectedly switch languages, poses a fundamental challenge to multilingual generation in LLMs. However, we still lack a mechanistic account of how this failure mode is implemented inside the model. Key questions remain: what internal components (i.e., *circuits*) give rise to unintended code-switching, where do they emerge across layers, and how can we intervene to mitigate it? In this work, we introduce a scalable circuit discovery framework that causally *localizes* multilingual neurons and *describes* their functional patterns, then *further groups* them into interpretable circuits—without any additional training or manual annotation. Our findings are twofold: a) The model's "speaking-a-language" circuit decomposes into a language regime (detecting and maintaining language identity) and a semantic regime (retrieving language-agnostic semantics). b) The mechanism of unintended code-switching is a regime shift. The semantic regime suppresses the language regime and overwhelms the multilingual circuit, causing the model to generate in an unintended language. To validate these findings, we further fine-tune the identified language sub-circuit, reducing the code-switching rate by $20.8\%$ with minimal parameter updates ($\sim 0.019\%$ of all neurons). This work serves as a preliminary exploration of multilingual generation mechanism, offering actionable insight for targeted training for multilingual LLMs.

---

*Work done during Yuxin Xiao's research internship at Alibaba Cloud.

[1]State Key Lab of CAD&CG, Zhejiang University [2]Alibaba Cloud [3]School of Software Technology, Zhejiang University [4]UniTTEC Co.,Ltd [5]Beijing Automobile Works Co.,Ltd [6]Fabu Inc.. Correspondence to: Wenxiao Wang <wenxiaowang@zju.edu.cn>, Xu Shen <shenxu.sx@alibaba-inc.com>.

*Proceedings of the 43rd International Conference on Machine Learning*, Seoul, South Korea. PMLR 306, 2026. Copyright 2026 by the author(s).

## 1. Introduction

Large Language Models (LLMs) exhibit strong multilingual abilities in text understanding and generation (Alec Radford & Sutskever, 2019; Hoffmann et al., 2022; Huang et al., 2023; Zhang et al., 2023; Zhao et al., 2024a). Yet, recent studies reveal unintended code-switching[1]—mixing languages within a single utterance—during generation (DeepSeek-AI et al., 2025; Dubey et al., 2024; Lu et al., 2024). For example: "Stephen Surjik est le réalisateur principal de ce film de 恐怖片" (English: Stephen Surjik is the lead director of this horror movie), a case generated by Qwen2.5-7B-Instruct model. Here, the French word peur ("horror") is incorrectly replaced by the Chinese 恐怖, yielding an unnatural switch.

While code-switching is natural in human multilingual communication (Auer & Wei, 2007; Gumperz, 1982), model-generated switches often violate linguistic constraints and appear unpredictable. Despite prior work using language-related neurons/features to mitigate unintended code-switching (Nie et al., 2025; Deng et al., 2025), it remains unclear how language selection is causally implemented during generation and why it sometimes fails and sometimes succeeds.

In this work, we aim to seek a mechanistic understanding of how language selection operates during multilingual generation. First, we view multilingual generation as the result of multiple internal components (*e.g.*, attention heads and MLP neurons) jointly steering the next-token distribution. During generation, these components usually keep the model in the intended language. Therefore, to understand unintended code-switching, we must identify how these components are organized, how information flows between them, and when this control breaks down. We study three important yet unanswered research questions:

**RQ1**. What is the internal architecture that supports multilingual language generation in LLMs?

**RQ2**. What internal mechanism leads to unintended code-switching during multilingual generation?

---

[1]Also known as *language confusion* (Marchisio et al., 2024)

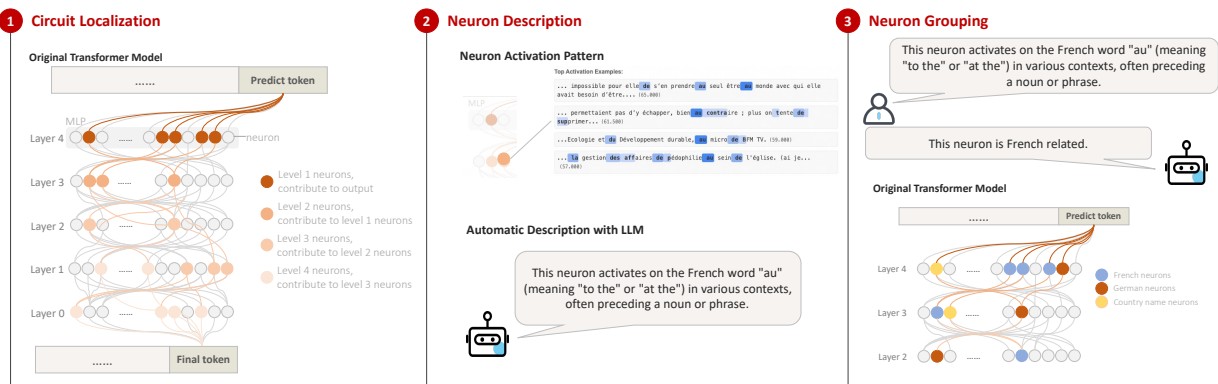

*Figure 1.* Overview of the proposed circuit discovery framework for interpreting unintended code-switching. Given an unmodified transformer model, we (1) discover circuits composed of hierarchical MLP neurons to explain their underlying mechanisms. (2) identify neurons' top activation samples, then employ an explainer model to generate textual descriptions of activation patterns, and (3) cluster functionally similar neurons. This structured grouping enhances circuit interpretability by consolidating semantically aligned components.

**RQ3**. Does understanding the underlying mechanism of unintended code-switching enable effective mitigation?

To study these hypotheses mechanistically, we introduce a scalable causal circuit discovery framework (Figure 1). Our key technical step is to extend attribution patching (Nanda, 2022) beyond tracing component-to-output effects. This extension allows us to trace component-to-component attributions and enables end-to-end circuit extraction. Then, these localized neurons are annotated and clustered into super-neurons for interpretability, leading to a circuit of multilingual generation.

The discovered circuit comprises two functional components. The first consists of language-specific super-neurons, which track contextual language identity. The second consists of semantic super-neurons, which encode language-agnostic content (*e.g.*, horror-related concepts). Figure 2a illustrates this structure in a case of normal French generation, where the model correctly completes with "peur" (horror). In this setting, activation follows a *two-step* pattern, in which semantic super-neurons influence the final prediction via language-specific super-neurons. Figures 2b and 2c present a different prompt that would otherwise lead to unintended code-switching. When semantic super-neurons exert a dominant influence on the output (Figure 2b), semantic signals bypass language selection, resulting in cross-lingual token generation. Conversely, by strengthening language-specific super-neurons, the model recovers the intended language behavior (Figure 2c). Additional examples of super-neurons are shown in Figure 4.

Building on these observations, we conduct a comprehensive experimental study of multilingual generation to address all three research questions. For **RQ1**, we identify language-related circuits across five models and eight lan-guages, and verify their causal necessity. Circuit-level interventions, clamping the activations of identified neurons, lead to severe degradation in multilingual generation performance, with drops of up to 97.3%. For **RQ2**, attribution-based analyses show that language-related circuits dominate the output distribution in non-code-switched cases ($2.37\times$ higher attribution than semantic). This dominance is substantially reduced in code-switched generations, revealing circuit-level competition and semantic bypass of language control. Finally, addressing **RQ3**, we evaluate mechanism-motivated mitigation across five task types, and show that fine-tuning only 0.019% of all neurons reduces unintended code-switching by 20.8% on average.

Our contributions are summarized as follows:

- We introduce a causal circuit discovery framework that reveals language circuits essential for maintaining language consistency in multilingual generation.
- We identify a circuit-level mechanism in which unintended code-switching arises from competition between language and semantic pathways.
- We show that this mechanistic understanding enables effective mitigation across diverse inputs by selectively fine-tuning a small subset of neurons.

**Conflict of Interest Disclosure.** Yuxin Xiao conducted this work during a research internship at Alibaba Cloud. Zhen Huang, Xu Shen, and Jieping Ye are employed by Alibaba Cloud. This paper evaluates Qwen2.5-7B-Instruct, a model developed by Alibaba Cloud; the Alibaba-affiliated authors were not involved in the development of this model.

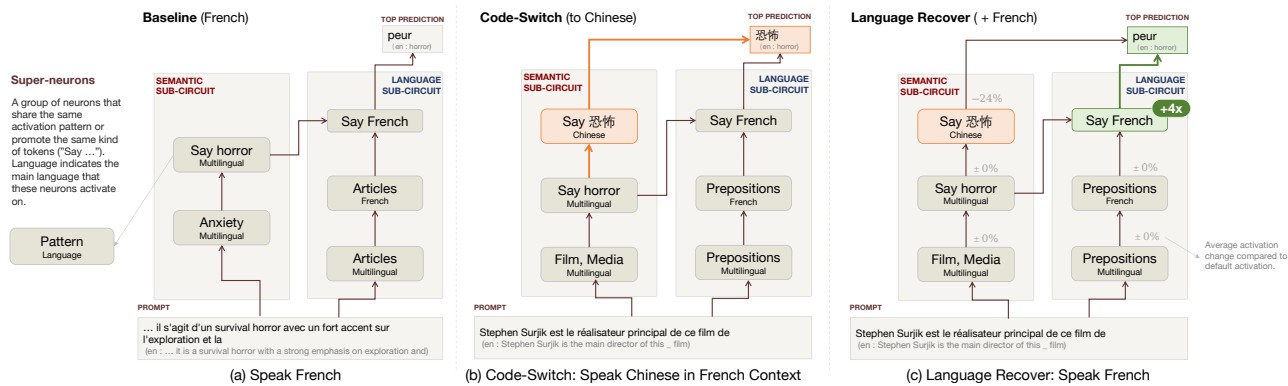

*Figure 2.* Circuits for: (a) normal French generation; (b) code-switching (Chinese tokens in a French context); and (c) recovery via activation manipulation. Lighter background boxes indicate semantic (left) and language (right) sub-circuits; Darker inner boxes denote super-neurons. Within each super-neuron, the upper (or middle) text represents activation patterns and the lower text indicates the dominant language. In normal French generation, semantic information flows from a language-agnostic semantic sub-circuit to a language-specific sub-circuit for final prediction. Code-switching emerges when semantic signals directly influence the output, bypassing language selection. Intervening on the language-indicating super-neuron suppresses code-switched tokens. Detailed circuits are shown in Fig. 11 and Fig. 12.

## 2. Related Works

Research on multilingualism in LLMs ranges from identifying language subspaces (Xie et al., 2024; Chang et al., 2022) to locating discrete language-specific neurons and features (Zhang et al., 2024c; Zhao et al., 2024b; Tang et al., 2024; Lindsey et al., 2025; Ameisen et al., 2025). These studies reveal where multilingual information may reside, but they generally lack causal validation of how such components drive cross-lingual behavior. Recent work further examines how language and semantic information are organized inside multilingual models. Wendler et al.(Wendler et al., 2024) propose a sequential transition from input-language representations to language-agnostic concepts and then to output-language representations. Dumas et al. (Dumas et al., 2025a) suggest that linguistic form and semantic content can be partially disentangled, while Ferrando et al. (Ferrando & Costa-jussà, 2024) analyze static information-flow routes for multilingual processing. Another work on language confusion (Nie et al., 2025) selects neurons by measuring log-probability shifts when injecting activations into the residual stream, showing that confusion-related neurons tend to appear in later layers. Our approach differs by using attribution-based neuron patching to recover circuits that causally influence both semantic and language-specific processing, highlighting competitive interactions rather than only confusion-related units.

More broadly, mechanistic interpretability has progressed from causal mediation analysis (Vig et al., 2020; Pearl, 2001) to circuit-level explanations via Path Patching (Olah et al., 2020; Wang et al., 2022; Hanna et al., 2023; Lieberum et al., 2023), though these methods are costly at scale. Attribution Patching (Nanda, 2022) and its efficient variants (Kramár et al., 2024; Syed et al., 2023) address this limita-

tion and have been applied to multilingual settings (Dumas et al., 2025b). Whereas (Dumas et al., 2025b) argues for a sequential "language-then-concept" mechanism using activation patching, our neuron-level attribution approach instead reveals parallel pathways, where competition between semantic and language circuits explains unintended code-switching—offering a complementary view of multilingual processing.

## 3. Preliminary

To analyze the mechanisms underlying multilingual generation, we adopt attribution patching (Nanda, 2022; Syed et al., 2023; Kramár et al., 2024) to approximate causal effects via linear interventions. While prior circuit analyses use attribution patching as a single-step tool that yields localized evidence (Marks et al., 2024), we extend it to iterative causal tracing by recursively patching connected neurons from the output back to the embedding layer, enabling reconstruction of neuron-level causal circuits.

**Attribution Patching.** Given an LLM $M$, a contrastive pair of inputs $(x_{clean}, x_{corrupted})$, and a metric $m$, let $n \in \mathbb{R}^d$ be a neuron [2] and $a \in \mathbb{R}$ its activation. Following (Nanda, 2022), the attribution of $n$ is:

$$\text{AttP}(m; a; x_{\text{clean}}, x_{\text{corrupted}}) = \nabla_a m\big|_{a=a_{\text{clean}}} \\ (a_{\text{clean}} - a_{\text{corrupted}}) \quad (1)$$

Here $a_{clean}$ is $n$'s activation on the last token given input $x_{clean}$, and $\nabla_a m|_{a=a_{clean}}$ represents the gradient of $a$ when running on $x_{corrupted}$ but *intervening* by manually setting

---

[2]Following (Geva et al., 2021b), "neuron" refers to a column of the MLP down-projection.

$a$ to $a_{clean}$. For example, given inputs $x_{clean}$="Angola is located in " and $x_{corrupted}$= "Angola liegt in " (German), we have metric

$$m(x) = \frac{LD_{patch} - LD_{corrupted}}{LD_{clean} - LD_{corrupted}} \quad (2)$$

where $LD = \text{Logit}[\texttt{Africa}] - \text{Logit}[\texttt{Afrika}]$ (Afrika is Africa in German), and $LD_{patch}$ refers to the logit difference when $a$ is patched to $a_{clean}$. Then a large value of $\text{AttP}(m; a; x_{clean}, x_{corrupted})$ indicates that the neuron is highly influential on the model's decision to output Africa rather than Afrika on this pair of inputs. We provide a detailed derivation for Equation 1 in Appendix A.1.4.

Attribution patching efficiently identifies neurons contributing to the output with only two forward passes and one backward pass under linearity assumptions, but it does not reveal the upstream causes of these contributions. To address this, we propose *hierarchical attribution patching*, which traces upstream neurons of $n$ by measuring the joint effect of paths involving both the upstream neuron and $n$.

## 4. Method

In this section, we present a general framework for uncovering the mechanisms behind code-switching in multilingual LLMs. The framework consists of three stages: *Circuit Localization*, *Neuron Description*, and *Neuron Grouping*. Section 4.1 describes how we identify language-related neuron circuits, while Sections 4.2 and 4.3 explain how we automatically describe and group the localized neurons for scalable interpretation. We use "circuit" to denote a set of causally connected MLP neurons identified by hierarchical attribution patching, and "super-neuron" to denote an interpretable group of neurons with similar activation descriptions and functional roles. An overview of the full pipeline is shown in Figure 1.

### 4.1. Circuit Localization

**Hierarchical Attribution Patching.** The core intuition behind hierarchical attribution patching is to conceptualize a model as a computational graph, where each neuron functions as a component that reads from and writes to the residual stream (Nanda, 2022; Geva et al., 2022). Each late neuron aggregates inputs from earlier nodes, and the residual stream aggregates the outputs of all preceding components; to isolate one edge, we patch only the early neuron's output into the late neuron's input while freezing others.

Specifically, let $n_e$ denote the early neuron for a late neuron $n_l$, where the layer index $l_e$ for $n_e$ is shallower than $l_l$, their edge effect can be calculated as ($x_{clean}$ and $x_{corrupted}$ are abbreviated as $x_{cle}$ and $x_{cor}$, respectively; the same applies

to $a_{clean}$ and $a_{corrupted}$. ):

$$\text{AttP}_{edge}(m; a_e, a_l; x_{cle}, x_{cor}) = \nabla_{a_l} m\big|_{a_l = a_{l,cle}} \quad (3)$$
$$(a_{e,cle} - a_{e,cor})$$

where every value is projected onto the residual stream to maintain linearity and additivity.

We showcase the pseudo-code of our method in Algorithm 1, and a detailed mathematical derivation of Equation 3 in Appendix A.1.4. For the early node, we obtain its patched output in the residual stream by multiplying the scalar activation $n_e$ by the corresponding row of the MLP down-projection matrix. For the late node, we restrict computation to the edge mediated by the up-projection matrix rather than the gated projection, as the latter (e.g., via SiLU activation) would compromise linearity. To compute the gradient of the metric $m$ with respect to the residual stream—mediated solely by $n_l$, we first remove the SiLU output in $n_l$ and then multiply by the relevant row of the up-projection matrix, yielding $\nabla_{a_l} m\big|_{a_l = a_{l,clean}}$.

In practice, we identify "level-1" neurons directly influencing the output, then iteratively trace upstream neurons whose edge attribution exceeds $\epsilon$. This continues until either reaching the embedding layer or a maximum depth $L$ (set to 5 with $\epsilon = 0.001$), producing hierarchical circuits. We provide ablation studies on $L$ and $\epsilon$ in Appendix A.1.6.

### 4.2. Neuron Description

Current neuron interpretation methods (Lee et al., 2024; Geva et al., 2022) primarily analyze neuron projection patterns, assuming each neuron promotes or suppresses token likelihoods via $P_{vocab}(n_{l,i}) = W_{out} \cdot W_{l,i}$, where $W_{out}$ is the output embedding matrix and $W_{l,i}$ the weights of neuron $n_{l,i}$. However, neurons often exhibit superposition, activating for multiple concepts across inputs, which static projections alone cannot fully capture. Prior work shows that incorporating activation patterns can enrich projection-based analysis by revealing more interpretable neuron behavior (Choi et al., 2024). From a key–value memory perspective of feed-forward layers (Geva et al., 2021b), projections reflect how neurons influence outputs, while activations capture the input patterns that trigger them, motivating a combined view of neuron function.

Accordingly, we rank tokens promoted by a neuron $n$ via dot products with unembedding matrix $e$, then analyze activations across a multilingual corpus. Sentences are sorted by maximum token activation $T_{max}(n_{l,i}) = t_j | \text{activation}(n_{l,i}, t_j) > \alpha$, where $\text{activation}(n_{l,i}, t_j)$ denotes the activation of neuron $n_{l,i}$ on the $t_j$-th token of the input, producing a top-$k$ set representative of $n$'s behavior. Integrating projection and activation patterns, we construct neuron profiles with LLMs: $P(n_{l,i}) =$

$LLM(P_{\text{vocab}}(n_{l,i}), T_{\max}(n_{l,i}))$. This reveals the contexts that trigger $n$, the semantics it encodes, and the tokens it promotes. Implementation details are provided in Appendix A.1.3; neuron description validation experiments are provided in Appendix A.3.4; and the prompts used for neuron description are provided in Appendix A.4.

## 4.3. Neuron Grouping

To better interpret the circuit, we group neurons with related semantics and functions into super-neurons, following (Lindsey et al., 2025). This yields a simplified view of the model's computation. Inspired by clustering in machine learning (Ester et al., 1998), we pre-define categories such as language-specific (e.g., "German," "French") and context-specific (e.g., "Media," "Horror"). Each neuron's description from Section 4.2 is then classified by a judge model: $C(n_{l,i}) = LLM(P(n_{l,i}), c_1, c_2, \ldots, c_k)$, where $C(n_{l,i})$ is the assigned category and $c_1, \ldots, c_k$ are existing classes. If no match is found, a new category is generated and added to the taxonomy, producing a dictionary that maps labels to neuron groups. We provide the grouping validation results in Appendix A.3.4 and grouping prompt in Appendix A.4.

# 5. Experiments

Our experiments are divided into five parts: circuit discovery and evaluation (Section 5.1), circuit validation (Section 5.2), discussion of code-switching through circuit competition (Section 5.3), fine-tuning neurons (Section 5.4), and an analysis of how input languages and tasks influence model's code-switching performance (Section 5.5). We conduct our experiments on Qwen2.5-7B-Instruct (Yang et al., 2024) and LLaMA3.1-8B-Instruct (Dubey et al., 2024) to validate the scalability of our method. We ran all experiments using four 80GB A100 GPUs. We focus on French, Spanish, Russian, Chinese-simplified, Arabic, Japanese, Vietnamese, and Indonesian for most experiments. The nuances of our language selection, hyper-parameter setting, dataset overview, and patching data construction are clarified in Appendix A.1.8, A.1.6, A.2.1, and A.2.2.

## 5.1. Circuit Discovery and Evaluation

To address **RQ1**, we characterize the internal architecture that supports multilingual language generation by analyzing the last-token MLP neurons, which directly influence language selection (Geva et al., 2023; Zhao et al., 2024b). We examine whether multilingual behavior can be explained by a structured and compact set of internal components rather than being diffusely distributed across the network.

**Circuit Discovery Details.** Within the identified architecture, approximately 30% of neurons exhibit activation patterns across multiple languages, consistent with

prior findings on multilingual core neurons (Zhang et al., 2024c), while 60–70% remain monolingual. Notably, this proportion is stable across both code-switched and non-code-switched samples, suggesting that unintended code-switching is not driven by excessive multilingual interference alone (Appendix A.3.2).

**Circuit Evaluation.** We construct circuits for all eight languages and evaluate them using two metrics from (Marks et al., 2024): faithfulness and completeness. Given a circuit $C$ and metric $m$, let $m(C)$ denote the average value of $m$ over a test dataset $D$ when running our model with all neurons outside of $C$ mean-ablated, i.e., set to their average value over data at each token from $D$. Let $m(\phi)$ denote the score when all neurons in the circuit are zero-ablated, and $m(M)$ the score of the full model with no intervention. We then measure faithfulness as $\frac{m(C)-m(\emptyset)}{m(M)-m(\emptyset)}$, which quantifies how much of the model's performance is captured by the circuit. Completeness is computed in the same way but using the circuit's complement $M \setminus C$, indicating how much behavior the circuit fails to account for. Thus, lower completeness scores indicate that little task-relevant behavior remains after removing the identified circuit.

As shown in Figure 3, our results reveal that small circuits of fewer than 50 neurons account for most task performance (faithfulness), while ablating even a subset of these neurons leads to substantial performance degradation (completeness). This behavior indicates that the identified circuits function as minimal, non-redundant units. We further validate the effectiveness of our circuit discovery framework on additional model abilities in Appendix A.3.11.

## 5.2. Language Neurons Validation for Multilingual Generation Tasks

Multilingual generation requires the model to maintain a target language consistently throughout decoding. If language control were distributed across many redundant components, suppressing a small subset of language-related neurons would be expected to cause only limited and non-specific degradation. In this section, we examine whether the language-related components identified in Section 5.1 constitute causally necessary parts of the internal architecture for preserving language identity.

We apply our method to Qwen2.5-7B-Instruct and LLaMA3.1-8B-Instruct to obtain language circuits for 8 languages, and validate them by clamping all circuit neurons to negative values at every token. Intervention strengths are selected empirically, following standard circuit-analysis practice, to test whether the identified neurons exert consistent causal influence on multilingual generation; sensitivity analyses are in Appendix A.1.7.

Following (Zhao et al., 2024b), we evaluate multilingual

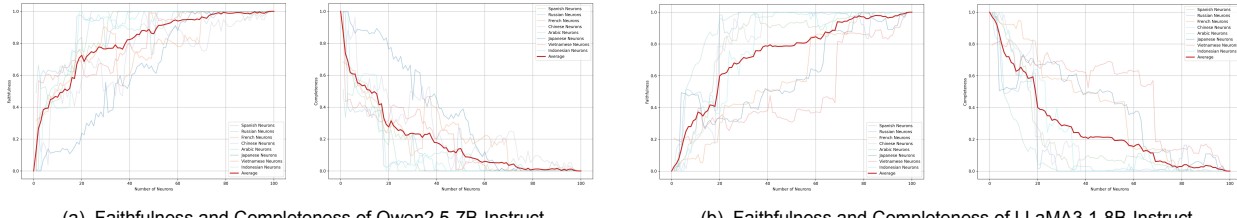

(a). Faithfulness and Completeness of Qwen2.5-7B-Instruct      (b). Faithfulness and Completeness of LLaMA3.1-8B-Instruct

*Figure 3.* Faithfulness and completeness of Qwen2.5-7B-Instruct (a) and LLaMA3.1-8B-Instruct (b) measured on $D$. Faint lines correspond to the different language circuits, with the average in bold and red. The ideal faithfulness for circuits is 1, while the ideal scores for their completeness is 0.

performance on XLSum (Hasan et al., 2021), which requires both comprehension and generation in the target language. We compare performance after deactivating language-related neurons against a random-ablation control with the same number of neurons. Additional results on Mistral-7B-Instruct-v0.3, Gemma2-9B-Instruct, and Qwen2.5-32B-Instruct are in Appendix A.3.1.

Table 1 shows that deactivating language-specific neurons selectively degrades performance in the corresponding language, where $D_s$ denotes the performance decline in the suppressed language and $D_{\text{non-s}}$ the average decline in other languages. We quantify language specificity using $\Delta = D_s - D_{\text{non-s}}$, with larger values indicating stronger language specialization. Notably, deactivating only $0.018\%$ of neurons causes severe and selective performance drops (e.g., up to $97.3\%$ for French in LLaMA and $93.9\%$ for Chinese in Qwen), while other languages and random neuron ablations are largely unaffected.

Together, these results show that language-related components form a small but causally necessary part of the architecture for multilingual generation and operate independently across languages. This finding makes direct competition between language-specific circuits an unlikely explanation, and motivates a closer examination of interactions between language control and other internal processes.

### 5.3. Dynamics of Unintended Code-Switching

To address **RQ2**, we examine how unintended code-switching emerges during multilingual generation. Unlike linguistically constrained phenomena, code-switching does not occur at fixed or predefined positions in the output. Instead, it can arise at arbitrary points during decoding, with its likelihood increasing as generation proceeds and outputs become longer and less constrained. This pattern suggests that code-switching is not caused by a single localized failure, but reflects **a gradual change in the internal dynamics that govern generation over time**.

To characterize this change, we analyze how the relative influence of internal components on the output evolves during

decoding. Attribution provides a natural tool for this purpose, as it quantifies each component's contribution through its effect on output logits (Ameisen et al., 2025; Nanda, 2022). We compare code-switched and non-code-switched generations using circuit-level attribution. Specifically, we sample 10 code-switched and 10 non-code-switched examples per language, construct three-level circuits, and group super-neurons based on their functional profiles. We then examine their relative attribution strengths at the token immediately preceding a code-switching event, or at the corresponding position in non-code-switched generations.

Averaged attribution ratios (Table 2) are computed by dividing the mean attribution of language neurons by that of semantic neurons. In non-code-switched cases, language circuits clearly dominate, with an average ratio of $2.37\times$, whereas this dominance is substantially reduced in code-switched cases, where the ratio drops to $1.21\times$. Notably, the ratio remains above 1.0 in most cases, suggesting these are "potential" switch points where the code-switched token is probable but not yet top-ranked. Together, these results show that code-switching reflects a weakened dominance of language circuits and increased competition from semantic circuits. A mathematical derivation of this competence is provided in Appendix A.1.5.

To directly validate the causal role of semantic circuits, we suppress their activations at the token preceding the confusion point in code-switched samples, and compare the model's prediction probabilities for the original code-switched token and an alternative token aligned with the intended language, generated with temperature 0.0. As shown in Table 3, this intervention reduces the probability of the code-switched token by $99.67\%$ ($0.06 \rightarrow 0.0002$) while increasing the probability of the intended-language token by $67.68\%$ ($0.3964 \rightarrow 0.6647$). These changes indicate that **unintended code-switching arises when non-language-specific components exert increased influence on the output distribution**, weakening language control and leading to circuit-level interference during decoding, thereby addressing **RQ2**. Crucially, this sharp reversal through minimal intervention suggests that such imbalances may be

*Table 1.* Multilingual performance on XLSum when deactivating language-specific neurons in the circuit (Fr-neurons, Zh-neurons, Es-neurons, Ru-neurons, Ar-neurons, and Ja-neurons) and an equivalent number of randomly selected neurons (Random). Deactivating $N_l$ selectively degrades performance in language $l$, while other languages are largely preserved. We use $N_l$ to denote the neurons corresponding to the language $l$, whereas random neurons are represented as $N_{rand}$.

| Deactivated | Performances Across Languages (%) | | | | | | | | | | |
|---|---|---|---|---|---|---|---|---|---|---|---|
| Neurons | Fr | Zh | Es | Ru | Ar | Ja | Vi | Id | $D_{\mathrm{s}}$ | $D_{\mathrm{non-s}}$ | $\Delta\ (\uparrow)$ |
| *Qwen2.5-7B-Instruct* | | | | | | | | | | | |
| None | 23.46 | 24.75 | 18.77 | 22.62 | 24.53 | 31.42 | 23.85 | 25.17 | - | - | - |
| $N_{rand}$ | 23.77 | 23.67 | 18.45 | 22.14 | 25.19 | 30.82 | 23.40 | 24.60 | - | - | - |
| $N_{Fr}$ | 6.53 | 23.66 | 17.58 | 18.40 | 23.16 | 27.90 | 17.72 | 22.09 | 16.93 | 1.08 | 15.85 |
| $N_{Zh}$ | 23.64 | 1.51 | 18.84 | 18.34 | 23.43 | 29.01 | 19.37 | 24.06 | 23.24 | 1.44 | 21.80 |
| $N_{Es}$ | 21.90 | 23.26 | 8.13 | 21.70 | 20.42 | 28.65 | 19.47 | 24.00 | 10.64 | 0.65 | 9.99 |
| $N_{Ru}$ | 22.64 | 23.27 | 18.47 | 3.87 | 23.59 | 30.22 | 20.37 | 19.02 | 18.75 | 0.95 | 17.80 |
| $N_{Ar}$ | 23.57 | 23.76 | 18.56 | 22.47 | 8.20 | 30.21 | 23.42 | 24.60 | 16.33 | 1.85 | 15.84 |
| $N_{Ja}$ | 22.13 | 22.18 | 16.81 | 21.04 | 22.70 | 2.58 | 21.68 | 24.35 | 28.84 | 1.01 | 27.83 |
| $N_{Vi}$ | 22.89 | 24.06 | 18.15 | 22.41 | 23.15 | 30.45 | 3.48 | 24.33 | 20.37 | 0.24 | 20.13 |
| $N_{Id}$ | 23.52 | 23.61 | 18.61 | 22.24 | 24.23 | 31.15 | 23.82 | 8.99 | 16.18 | 2.25 | 13.93 |
| *LLaMA3.1-8B-Instruct* | | | | | | | | | | | |
| None | 24.24 | 29.70 | 20.97 | 25.15 | 25.46 | 33.11 | 20.86 | 25.61 | - | - | - |
| $N_{rand}$ | 24.89 | 29.82 | 20.73 | 24.10 | 26.68 | 33.55 | 20.89 | 25.71 | - | - | - |
| $N_{Fr}$ | 0.66 | 29.56 | 19.87 | 25.05 | 25.12 | 32.92 | 17.95 | 21.56 | 23.58 | 1.07 | 22.51 |
| $N_{Zh}$ | 22.39 | 1.50 | 19.53 | 24.45 | 24.38 | 32.57 | 22.60 | 24.18 | 28.20 | 0.33 | 27.87 |
| $N_{Es}$ | 23.21 | 23.55 | 2.73 | 23.07 | 24.38 | 31.69 | 19.29 | 21.55 | 18.24 | 1.24 | 17.00 |
| $N_{Ru}$ | 23.71 | 29.12 | 19.80 | 6.20 | 24.78 | 33.00 | 20.56 | 25.75 | 18.95 | 1.31 | 17.64 |
| $N_{Ar}$ | 22.78 | 26.81 | 19.37 | 24.94 | 2.05 | 32.89 | 21.28 | 27.08 | 23.41 | 0.06 | 23.35 |
| $N_{Ja}$ | 23.67 | 28.36 | 19.77 | 24.83 | 23.50 | 4.34 | 23.49 | 26.69 | 28.77 | -1.00 | 27.77 |
| $N_{Vi}$ | 23.47 | 28.18 | 21.53 | 23.69 | 24.66 | 30.58 | 7.41 | 24.14 | 13.45 | 0.98 | 12.47 |
| $N_{Id}$ | 24.01 | 29.26 | 20.18 | 23.93 | 26.38 | 32.99 | 21.25 | 5.13 | 20.48 | 1.25 | 19.23 |

corrected by precisely modulating a small set of components, motivating the mitigation strategies explored next.

We additionally analyze layer-wise behavior using Log-itLens (nostalgebraist, 2020) (Appendix A.3.3) and observe a distinct pattern: in normal generation, probabilities of intermediate semantic tokens decay sharply in later layers, whereas in code-switched samples they remain elevated, indicating insufficient suppression before the final output.

## 5.4. Code-Switch Evaluation and Reduction

Building on the mechanism identified in Section 5.3, we first assess how frequently unintended code-switching occurs in practice, and then examine whether this understanding can inform effective mitigation. We evaluate models using the Language Confusion Benchmark (LCB) (Nie et al., 2025), which extends monolingual prompt settings from (Tramm et al., 2024) across a diverse set of languages, and measure performance with Line-level Pass Rate (LPR), defined as the percentage of responses in which all lines remain in the target language. Following the original setup, we use temperature = 0.3 and top_p = 0.75, regenerate each response 10 times, and increase the maximum output length to 1024 tokens to capture longer generations where code-switching is more likely (see Appendix A.3.7 for additional analysis).

Experiments on Qwen2.5-7B-Instruct and LLaMA3.1-8B-Instruct (Table 4, first column under each model's name) show that unintended code-switching remains non-trivial across languages and tasks. LPR is a strict metric, where any foreign-language token leads to failure, making even small LPR differences practically meaningful.

Building on this quantitative characterization, we next examine whether the mechanism identified in Section 5.3 can be leveraged to mitigate code-switching through targeted fine-tuning. Given the identified language-specific neuron list, we freeze all other model parameters and update only the corresponding columns in the MLP down-projection matrices. For each language, we train on monolingual samples from the Aya dataset. We fine-tune the selected neurons with a polynomial learning-rate schedule, maximum learning rate $5 \times 10^{-5}$, minimum learning rate $1 \times 10^{-6}$, weight decay 0.1, and batch size 64. Additional details on data construction and hyperparameters are provided in Appendix A.2 and Appendix A.1.6. Notably, language-specific neurons account for only about 0.019% of all neurons, so tuning them incurs minimal computational cost while preserving overall task performance. Details of training data construction are provided in Appendix A.2. For comparison, we also perform full-parameter fine-tuning using the same training

*Table 2.* Language-to-semantic attribution ratios for non-code-switched and code-switched circuits. NCSW denotes non-code-switched samples, CSW denotes code-switched samples, and each value is the attribution ratio between language-specific circuits and semantic circuits. A lower ratio in CSW indicates weaker dominance of language circuits and stronger semantic competition.

| Model | NCSW | CSW |
|---|---|---|
| Qwen2.5-7B-Instruct | 2.3717 | 1.2073 |
| LLaMA3.1-8B-Instruct | 3.0952 | 1.0083 |
| Mistral-7B-Instruct-v0.3 | 2.4897 | 1.4098 |

*Table 3.* Model's prediction probabilities for the original code-switched token and a non-code-switched token (generated with temperature 0.0 and aligned with the intended language), before and after intervention.

| Type | $\bar{P}$ before | $\bar{P}$ after |
|---|---|---|
| CSW token | 0.0608 | 0.0002 |
| NCSW token | 0.3964 | 0.6647 |

*Table 4.* LPR for the baseline model, fine-tuning random neurons ("$N_{Rand}$"), and fine-tuning language-specific neurons ("$N_{Lang-Spec}$").

| Trained Neurons | Line-level Pass Rate (↑) | | | | | | | | |
|---|---|---|---|---|---|---|---|---|---|
| | **Ar** | **Es** | **Fr** | **Ja** | **Ru** | **Zh** | **Vi** | **Id** | **Average** |
| *Qwen2.5-7B-Instruct* | | | | | | | | | |
| Baseline | 96.41% | 96.67% | 95.18% | 88.78% | 98.08% | 92.10% | 97.38% | 82.96% | 93.45% |
| $N_{Rand}$ | 91.87% | 96.13% | 94.25% | 87.70% | 95.66% | 93.65% | 98.20% | 86.70% | 93.02% |
| $N_{Lang-Spec}$ | **98.50%** | **97.56%** | **96.62%** | **98.49%** | **99.16%** | **96.35%** | **99.10%** | **92.60%** | **97.30%** |
| *LLaMA3.1-8B-Instruct* | | | | | | | | | |
| Baseline | 99.25% | 97.63% | 97.86% | 96.00% | 98.40% | 94.55% | 99.70% | 83.50% | 95.99% |
| $N_{Rand}$ | 99.37% | 96.63% | 97.97% | 95.70% | 99.40% | 93.30% | 99.80% | 84.20% | 95.83% |
| $N_{Lang-Spec}$ | **99.87%** | **98.50%** | **98.63%** | **97.80%** | **99.87%** | **96.55%** | **99.90%** | **88.20%** | **97.36%** |

data, with results reported in Appendix A.3.6.

The fine-tuning results are shown in Table 4 (detailed in Appendix A.3.5). While fine-tuning randomly selected neurons yields little improvement, selectively updating language-specific neurons consistently increases LPR across languages and models, raising average performance from $93.45\% \rightarrow 97.30\%$ on Qwen and from $95.99\% \rightarrow 97.36\%$ on LLaMA. These gains are achieved by updating only a few hundred neurons, highlighting the precision of the identified language-specific components. Comparisons with standard baselines, including few-shot prompting, full-parameter SFT and DPO, are provided in Appendix A.3.8. Beyond the main LCB results, we further examine the robustness and practicality of targeted mitigation from three perspectives. Precise tuning preserves the models' general capabilities on MMMLU and FLORES, as shown in Appendix A.3.9. Neuron-weight merging enables a unified mitigation model across all eight languages, with results in Appendix A.3.10. Finally, the mitigation effect remains stable across different decoding strategies, as reported in Appendix A.3.7.

Together, these results demonstrate that a mechanistic understanding of unintended code-switching can be directly leveraged to design effective and targeted mitigation strategies, answering **RQ3**.

## 5.5. Robustness of Code-Switching Mitigation Across Tasks and Languages

To further address **RQ3**, we examine whether the effectiveness of mechanism-motivated mitigation generalizes across input types and languages. Models may remain language-consistent in constrained tasks but exhibit increased confusion in more open-ended settings. We therefore analyze Arabic and Japanese as representative cases that stress different aspects of language control, reflecting linguistic contact and typological distance, respectively.

We evaluate baseline code-switching behavior of Qwen and LLaMA on Arabic and Japanese tasks spanning **translation** (Costa-jussà et al., 2022), **instruction following** (Zhang et al., 2024b), **QA** (Longpre et al., 2021; So et al., 2022; Artetxe et al., 2020), and **conversation** (Ding et al., 2023). Dataset details are provided in Appendix A.1.9, with additional real-world evaluations in Appendix A.3.12.

**Evaluation Method**: We evaluate using *Language Consistency Rate* (LCR) — the proportion of model outputs that remain entirely in the intended language without unintended code-switching. (details in Appendix A.1.9). We employ generation parameters (top_p=0.8, top_k=20, temperature=0.7, presence_penalty=1.5) to balance output diversity with a minimal code-switching rate and generate 128 responses for each question to compute the final LCR.

*Table 5.* Model's LCR when fine-tuning different types of neurons. A higher LCR indicates less unintended code-switching.

| Model | Language | Methods | LCR on Different Tasks (↑) | | | | |
|---|---|---|---|---|---|---|---|
| | | | Translation | IF | QA | Conversation | **Average** |
| **Qwen2.5-7B-Instruct** | **Ar** | None (Baseline) | 47.93% | 65.00% | 54.19% | 54.37% | 53.94% |
| | | $N_{Ar}$ | **67.23%** | **77.37%** | **81.86%** | **74.46%** | **75.23%** |
| | **Ja** | None (Baseline) | 74.95% | 49.31% | 50.53% | 32.48% | 51.82% |
| | | $N_{Ja}$ | **91.69%** | **75.55%** | **99.52%** | **62.11%** | **82.22%** |
| **LLaMA3.1-8B-Instruct** | **Ar** | None (Baseline) | 88.57% | 74.75% | 92.67% | 59.99% | 79.00% |
| | | $N_{Ar}$ | **95.97%** | **97.18%** | **95.04%** | **96.06%** | **96.06%** |
| | **Ja** | None (Baseline) | 89.49% | 61.43% | 79.90% | 46.95% | 69.44% |
| | | $N_{Ja}$ | **97.25%** | **89.05%** | **99.68%** | **87.71%** | **93.42%** |

**Results**: Table 5 demonstrates that targeted fine-tuning of just 50 Arabic-specific neurons increases overall LCR by 21.29%, confirming both the effectiveness and specificity of our identified circuit. From the results, we can conclude some observations:

- **Language Differences**: Despite being a low-resource language, Arabic generally causes lower code-switching (higher LCR) than Japanese. However, LCR varies significantly across tasks for both languages.
- **Task Differences**: Task type strongly affects LCR. Translation and QA typically yield higher LCR, likely because they produce shorter, finite responses with well-defined outputs. In contrast, Instruction Following and Conversation are open-ended and often result in longer outputs—conditions under which models are more prone to code-switching.

These findings show that both task structure and language significantly affect a model's tendency to code-switch, with longer and less constrained generation increasing the risk of unintended code-switching. Importantly, the consistent improvements observed across diverse tasks indicate that mechanism-motivated targeted fine-tuning remains effective beyond specific settings, further addressing **RQ3**. We additionally demonstrate in Appendix A.3.11 that this framework generalizes to other behaviors, including reasoning, sycophancy, and mathematical tasks.

## 6. Discussions

Several aspects of this work point to important directions for future research. First, our circuit discovery method relies on attribution-based approximations and focuses primarily on last-token MLP neurons. Although this design makes circuit discovery scalable, it may miss mechanisms involving attention heads, cross-token interactions, or longer-range generation dynamics. Extending the framework to these components is an important direction for future work.

Second, our functional interpretation depends on LLM-

assisted neuron descriptions and clustering. These descriptions provide useful high-level summaries of neuron behavior, but they may not fully capture all contexts in which a neuron is active. Future work could strengthen this step with broader human validation, task-specific probes, or more systematic tests of description faithfulness.

Third, measuring unintended code-switching remains challenging. Language-identification tools can be sensitive to named entities, borrowed words, and transliterations. Although our additional analyses show consistent trends across alternative settings, more robust multilingual evaluation protocols would further improve reliability.

Finally, targeted neuron-level interventions should be applied carefully, since the same mechanism could potentially be used for unintended model steering. A promising future direction is to study how multilingual control interacts with reasoning and other high-level capabilities, which may further improve cross-lingual generalization.

## 7. Conclusion

In this work, we present a mechanistic account of multilingual generation in LLMs through unintended code-switching. We show that multilingual generation relies on a compact internal architecture in which language-related components are causally necessary for maintaining language consistency (**RQ1**), and that unintended code-switching arises not from direct competition between languages but from increased influence of semantic pathways that weaken language control during generation (**RQ2**). Finally, we demonstrate that this mechanistic understanding enables effective and targeted mitigation, substantially reducing code-switching by selectively fine-tuning a small subset of language-specific neurons without degrading overall performance (**RQ3**). Together, these findings highlight the value of circuit-level analysis for explaining and mitigating failures in multilingual language generation.

# Acknowledgment

This work was supported in part by The National Nature Science Foundation of China (Grant NOs.: 62303406, 62273303, 62432014)in part by Yongjiang Talent Introduction Programme (2022A-240-G).

# Impact Statement

This paper presents work whose goal is to advance the field of machine learning through improved understanding of multilingual language models. While our findings may inform model training and robustness, we do not foresee societal consequences that require specific discussion beyond standard considerations for responsible deployment of large language models.

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

# A. Appendix

## A.1. Implementation Details

### A.1.1. EXAMPLES OF SUPER-NEURONS

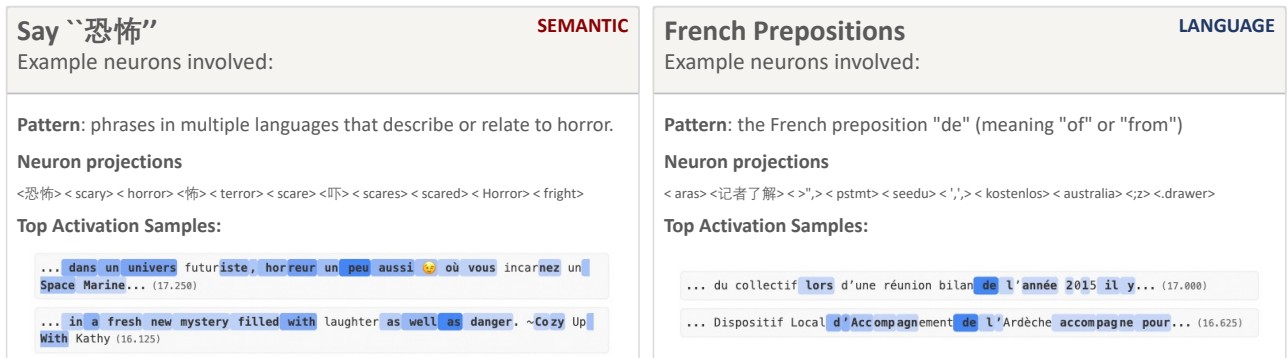

*Figure 4.* Example super-neurons and neurons which participate in the code-switching circuit. Neurons are active on tokens shaded in blue, where darker color indicates stronger activation value.

### A.1.2. ALGORITHM

Below we show the pseudo-code of our Hierarchical Attribution Patching Circuit Discovery method.

---

**Algorithm 1** Hierarchical Attribution Patching Circuit Discovery

---

**Require:** Model $M$, clean/corrupted data $(X_r, A_r)/(X_c, A_c)$, threshold $\epsilon$, max level $L$, attribution function $f$
**Ensure:** Hierarchical circuit neurons $\{C_1, \ldots, C_k\}$ where $k \leq L$

1: **function** MAIN$(M, X_r, X_c, A_r, A_c, \epsilon, L)$
2:    $C_1 \leftarrow \{(l, n) \mid f((l, n), \text{output}) > \epsilon\}$ {Level 1: output-attributing neurons}
3:    **for** $level \leftarrow 2$ **to** $L$ **do**
4:       $C_{level} \leftarrow \emptyset$
5:       **for** $(l_{end}, n_{end}) \in C_{level-1}$ **do**
6:          **for** $(l_{src}, n_{src})$ **with** $l_{src} < l_{end}$ **do**
7:             **if** $f((l_{src}, n_{src}), (l_{end}, n_{end})) > \epsilon$ **then**
8:                $C_{level} \leftarrow C_{level} \cup \{(l_{src}, n_{src})\}$
9:             **end if**
10:         **end for**
11:       **end for**
12:       **if** $C_{level} = \emptyset$ **then**
13:          **break**
14:       **end if**
15:    **end for**
16:    **return** $\{C_1, \ldots, C_{level}\}$
17: **end function**=0

---

### A.1.3. COMPUTATION DETAILS

In this section we provide details from (Geva et al., 2022) that demonstrate that MLP neurons promote or suppress the likelihood of tokens.

We start from Equation 4:

$$\text{MLP}^l(\mathbf{x}^l) = \sum_{i=1}^{d_{\text{MLP}}} \sigma(x^l \cdot \mathbf{k}_i^l)\mathbf{v}_i^l = \sum_{i=1}^{d_{\text{MLP}}} m_i^l \mathbf{v}_i^l \tag{4}$$

Thus, we can consider the update form $\text{MLP}^l$ as $d_{\text{MLP}}$ *sub-updates*, each sub-update being $m_i^l \mathbf{v}_i^l$.

We can then analyze the influence that each sub-update has on the output distribution, or the probability of generating token $\omega \in V$ (taken from (Geva et al., 2022)):

$$p(\omega|\mathbf{x}^l + m_i^l v_i^l, E) = \frac{\exp(\mathbf{e}_\omega \cdot \mathbf{x}^l + \mathbf{e}_\omega \cdot m_i^l \mathbf{v}_i^l)}{Z(E(\mathbf{x}^l + m_i^l \mathbf{v}_i^l))} \propto \exp(\mathbf{e}_\omega \cdot \mathbf{x}^l) \cdot \exp(\mathbf{e}_\omega \cdot m_i^l \mathbf{v}_i^l) \tag{5}$$

where $\mathbf{e}_\omega$ is the token embedding of $\omega$, and $Z$ is the softmax normalization factor. This indicates that when $\mathbf{e}_\omega \cdot m_i^l \mathbf{v}_i^l > 0$, the likelihood of $\omega$ increases, while $\mathbf{e}_\omega \cdot m_i^l \mathbf{v}_i^l < 0$ decreases the likelihood.

### A.1.4. DERIVATIONS OF EQUATION 1 AND 3

Let $i$ represent the index of a Chinese word in the vocabulary, and $j$ the index of the corresponding French word (e.g., at position $i$, we have "非洲" and at position $j$, "Afrika").

Define $e_i$ as the one-hot vector in $\mathbb{R}^V$ where the $i$-th position is 1 and the rest are 0. The logits corresponding to the Chinese and French tokens can then be written as:

$$\text{Logits}[非洲] = \text{Logits}^\top \cdot e_i, \quad \text{Logits}[\text{Afrika}] = \text{Logits}^\top \cdot e_j \tag{6}$$

Given a neuron with activation $a$, let $a_0 = a_{\text{clean}}$ represent the activation for a Chinese input, and $a_1 = a_{\text{corrupt}}$ represent the activation for a German input. Let $x_0$ denote the first $n-1$ tokens of the Chinese input and $x_1$ denote the first $n-1$ tokens of the German input.

$\text{Logits}(a, x)$ represents the logits produced by the model when the input is $x$ and the targeted neuron activation is forced to be $a$. So: - $\text{Logits}_{\text{clean}} = \text{Logits}(a_0, x_0)$ represents the model's original prediction on the clean Chinese input, using the neuron activation naturally induced by the Chinese context. - $\text{Logits}_{\text{corrupt}} = \text{Logits}(a_1, x_1)$ represents the model's prediction on the fully corrupted German input, using the neuron activation naturally induced by the German context. - $\text{Logits}_{\text{patch}} = \text{Logits}(a_0, x_1)$ represents a causal intervention where the German input is used but the neuron activation is overwritten with the activation from the Chinese instance, thereby isolating the causal effect of that neuron on the final logits.

Thus, the logit differences (LD) are:

$$\begin{aligned} LD_{\text{patch}} &= \text{Logits}_{\text{patch}}[非洲] - \text{Logits}_{\text{patch}}[\text{Afrika}] \\ &= \text{Logits}_{\text{patch}}^\top \cdot (e_i - e_j) = \text{Logits}(a_0, x_1)^\top \cdot (e_i - e_j) \\ LD_{\text{corrupt}} &= \text{Logits}(a_1, x_1)^\top \cdot (e_i - e_j) \\ LD_{\text{clean}} &= \text{Logits}(a_0, x_0)^\top \cdot (e_i - e_j) \end{aligned} \tag{7}$$

Given the formula above, the metric $m$ in Equation (2) can be written as:

$$m = \frac{(\text{Logits}(a_0, x_1) - \text{Logits}(a_1, x_1))^\top \cdot (e_i - e_j)}{(\text{Logits}(a_0, x_0) - \text{Logits}(a_1, x_1))^\top \cdot (e_i - e_j)} \tag{8}$$

**Deriving Equation (1)** : Think of $m$ as a function of the patched activation $m = f(a)$ under the corrupted input context. Holding all other activations fixed, we apply a first-order Taylor expansion around the patched clean activation $a_0$, so that

$$f(a) \approx f(a_0) + \nabla_a f(a_0)^\top \cdot (a - a_0) \tag{9}$$

Setting $a = a_1$ gives

$$\Delta m = f(a_0) - f(a_1) \approx \nabla_a f(a_0)^\top \cdot (a_0 - a_1) \tag{10}$$

This yields Equation (1):

$$AttP(m; a; x_0, x_1) = \nabla_a m|_{a=a_0}(a_0 - a_1) \tag{11}$$

This quantity measures the contribution of neuron activation $a$ to the final logits, i.e., the final node of the computation graph.

**Deriving Equation (3)**: Let $n_{l,i}$ be a neuron in layer $l$, indexed at $i$-th row in this layer's MLP down projection matrix. We can then denote early neuron and late neuron as $n_{l_1,i}$ and $n_{l_2,j}$ where $l_1 < l_2$.

Because of the linearity we assumed throughout the paper, we can consider the sum of the path attribution patch values over all start nodes (including the embedding) should equal the end node's total attribution patch value.

Thus:

$$a_{l_2,j} = a_{l_1,i} + \sum(\text{other residual terms}) \tag{12}$$

Although the activation values $a_{l_1,i}$ and $a_{l_2,j}$ typically pass through nonlinear activation functions (such as ReLU, GELU, etc.), the residual connection itself is based on a linear addition of information. This means that, even with nonlinear activation functions, the residual connection simply adds the output from earlier layers to the subsequent layers. Therefore, during backpropagation, the gradient flow through this linear addition is independent of the nonlinear activation functions. Specifically, the contribution of the early neuron to the late neuron is linear, as the gradient with respect to the early neuron's output is constant. This leads to the following relationship

$$\frac{\partial a_{l_2,j}}{\partial a_{l_1,i}} = I \tag{13}$$

This relationship holds for each neuron at the same level because the residual connection ensures that the influence of the early neuron on the late neuron remains linear, unaffected by the nonlinearities in intermediate layers.

Given the conclusion above, we can continue calculating the attribution of $n_{l_1,i}$ on $m$ through $n_{l_2,j}$. Since $m = f(a_{l_2,j}, \text{other residual outputs})$, and $a_{l_2,j}$ depends on $a_{l_1,i}$, we obtain

$$\frac{\partial m}{\partial a_{l_1,i}} = \frac{\partial m}{\partial a_{l_2,j}} \frac{\partial a_{l_2,j}}{\partial a_{l_1,i}} + \sum_{\text{other paths p}} \frac{\partial m}{\partial a_p} \frac{\partial a_p}{\partial a_{l_1,i}} \tag{14}$$

To isolate the attribution only along the path $n_{l_1,i} \to n_{l_2,j} \to m$, we retain the mediated term:

$$\frac{\partial m}{\partial a_{l_1,i}} = \frac{\partial m}{\partial a_{l_2,j}} \frac{\partial a_{l_2,j}}{\partial a_{l_1,i}} \tag{15}$$

Using $\frac{\partial a_{l_2,j}}{\partial a_{l_1,i}} = I$:

$$AttP_{n_{l_1,i} \to n_{l_2,j} \to m}(m; a_{l_1,i}, a_{l_2,j}; x_0, x_1) = \nabla_{a_{l_2,j}} m\big|_{a_{l_2,j}=a_{l_2,j,x_1}} (a_{l_1,i,x_0} - a_{l_1,i,x_1}) \tag{16}$$

which corresponds to the Equation (3):

$$AttP_{\text{edge}}(m; a_{l_1,i}, a_{l_2,j}; x_0, x_1) = \nabla_{a_{l_2,j}} m\big|_{a_{l_2,j}=a_{l_2,j,x_1}} (a_{l_1,i,x_0} - a_{l_1,i,x_1}) \tag{17}$$

A.1.5. DERIVATION OF COMPETENCE

Given Equation 8, we can treat $LD_{\text{corrupt}}$ and $LD_{\text{clean}}$ as constants when differentiating, hence we obtain:

$$\frac{\partial m}{\partial a}\bigg|_{a=a_0,x=x_1} = \frac{1}{(\text{Logits}(a_0, x_0) - \text{Logits}(a_1, x_1))^\top \cdot (e_i - e_j)} \left( \frac{\partial \text{Logits}(a, x)}{\partial a}\bigg|_{a=a_0,x=x_1} \right)^\top \cdot (e_i - e_j) \tag{18}$$

Hence, the attribution Attr is:

$$\text{Attr} = \frac{a_0 - a_1}{(\text{Logits}(a_0, x_0) - \text{Logits}(a_1, x_1))^\top \cdot (e_i - e_j)} \left( \left. \frac{\partial \text{Logits}(a, x)}{\partial a} \right|_{a=a_0, x=x_1} \right)^\top \cdot (e_i - e_j) \tag{19}$$

The activation $a$ represents the neural response to a specific input, which influences the output logits. Specifically, $\frac{\partial m}{\partial a}$ represents how much a change in the activation $a$ affects the difference between the logits for the Chinese and French tokens. This term measures the contribution of a particular neuron to the output, capturing the extent to which the neuron's activation influences the logit difference, or the "decision" for the model's output.

Now, let's introduce two types of neurons: one representing language-specific patterns whose activations are denoted by $a_l$, and the other representing semantic patterns whose activations are denoted by $a_s$. Notably, the denominator $(\text{Logits}(a_0, x_0) - \text{Logits}(a_1, x_1))^\top \cdot (e_i - e_j)$ is independent of which neuron we choose for patching (because it doesn't require patching and is calculated over fixed input), so it cancels out when comparing the ratio.

Thus, the ratio of attributions for the language and semantic neurons is:

$$\frac{\text{Attr}_l}{\text{Attr}_s} = \frac{(a_{l,0} - a_{l,1}) \left( \left. \frac{\partial \text{Logits}(a_l, x)}{\partial a_l} \right|_{a=a_{l,0}, x=x_1} \right)^\top \cdot (e_i - e_j)}{(a_{s,0} - a_{s,1}) \left( \left. \frac{\partial \text{Logits}(a_s, x)}{\partial a_s} \right|_{a=a_{s,0}, x=x_1} \right)^\top \cdot (e_i - e_j)} \tag{20}$$

To simplify, we can apply a first-order Taylor expansion around the numerator and denominator. For the language neurons and semantic neurons, we approximate:

$$\frac{Attr_l}{Attr_s} H \frac{(Logits(a_{l,0}, x_1) - Logits(a_{l,1}, x_1))^\top \cdot (e_i - e_j)}{(Logits(a_{s,0}, x_0) - Logits(a_{s,1}, x_1))^\top \cdot (e_i - e_j)}$$
$$= \frac{(Logits_{\text{patch-language}}[\text{非洲}] - Logits_{\text{corrupt}}[\text{非洲}]) - (Logits_{\text{patch-language}}[\text{Afrika}] - Logits_{\text{corrupt}}[\text{Afrika}])}{(Logits_{\text{patch-semantic}}[\text{非洲}] - Logits_{\text{corrupt}}[\text{非洲}]) - (Logits_{\text{patch-semantic}}[\text{Afrika}] - Logits_{\text{corrupt}}[\text{Afrika}])} \tag{21}$$

In **Non-Code-Switched Case**:

1. **Numerator**: When patching the language neuron, the Chinese token "非洲" exhibits a significant increase in probability, while the French token "Afrika" exhibits a significant decrease, resulting in a large positive numerator.
2. **Denominator**: After patching the semantic neuron, the probability for the Chinese token "非洲" should experience a slight increase, while the probability for the French token "Afrika" should experience a slight decrease. As a result, the first term in the denominator is a small positive value, and the second term is a small negative value, leading to a smaller positive denominator and a large attribution ratio.

In **Code-Switched Case**:

- **Numerator**: In the code-switched scenario, patching the language neuron still causes an increase for "非洲", but the decrease for "Afrika" is less pronounced. Consequently, the numerator is smaller compared to the normal case.
- **Denominator**: When patching the semantic neuron, its contribution no longer remains negligible relative to the language neuron. Instead, semantic attribution becomes comparable to language attribution because semantic pathways can directly influence the competing output tokens. As a result, the language-to-semantic attribution ratio decreases and approaches 1.

The experimental results in Section 5.3 align with this interpretation. In Table 2, the attribution ratio for non-code-switched samples is 2.37, dominated by language circuits, while for code-switched samples, the ratio decreases to 1.21, indicating a shift toward semantic circuits. Furthermore, as shown in Table 3, intervening on semantic circuits significantly reduced the probability of the target code-switched token, demonstrating that semantic circuits play a more prominent role in code-switching.

| $\epsilon$ | $L$ | Rouge Score ($\downarrow$) |
|---|---|---|
| 0.001 | 5 | **2.58** |
| 0.001 | 4 | 4.12 |
| 0.001 | 3 | 4.29 |
| 0.005 | 3 | 4.33 |
| 0.0001 | 3 | 3.97 |

*Table 6.* Japanese circuit

| $\epsilon$ | $L$ | Rouge Score ($\downarrow$) |
|---|---|---|
| 0.001 | 5 | **8.20** |
| 0.001 | 4 | 9.60 |
| 0.001 | 3 | 9.59 |
| 0.005 | 3 | 9.78 |
| 0.0001 | 3 | 9.26 |

*Table 7.* Arabic circuit

| $\epsilon$ | $L$ | Rouge Score ($\downarrow$) |
|---|---|---|
| 0.001 | 5 | **1.51** |
| 0.001 | 4 | 3.38 |
| 0.001 | 3 | 3.47 |
| 0.005 | 3 | 3.56 |
| 0.0001 | 3 | 3.28 |

*Table 8.* Chinese circuit

### A.1.6. HYPER-PARAMETERS

**Circuit Discovery Hyper-parameters.** We test various values of $\epsilon$ and $L$ in circuit discovery for Qwen2.5-7B-Instruct on Japanese, Arabic, and Chinese, and quantify the circuit precision using a suppression experiment, as shown in Table 1, where we applied a scale factor of -5 to the language-specific neurons' activations. The results are summarized in Table 6, Table 7, and Table 8.

From these experiments, we identify three main trends:

- Reducing $L$ (the number of levels and thus neurons) slightly increases ROUGE and speeds up both circuit discovery and neuron interpretation.
- Increasing $L$ increases computation time but generally improves precision.
- Increasing $\epsilon$ reduces the number of neurons while preserving key ones, resulting in minimal impact on final performance.

Based on this, we set $\epsilon = 0.001$ and $L = 5$ as a good balance between precision and efficiency, though these hyperparameters can be adjusted depending on the desired trade-off.

**Training Details.** We train 1000 neurons for each circuit without filtering out the multilingual neurons, but only based on their attribution to the next node in the circuit. Additionally, we set max learning rate=$5e-5$, min learning rate=$1e-6$, weight decay=$0.1$, learning scheduler type is set to polynomial, and batch size=$64$.

### A.1.7. CHOICE OF INTERVENTION STRENGTHS

Our choice of steering factors for intervention experiments is empirically guided rather than theoretically derived. For example, inhibition experiments often require clamping features to negative multiples of their original values—rather than zero—to meaningfully alter model outputs.

This observed need for "overcompensation" suggests our perturbation experiments only partially capture the underlying mechanisms, likely due to two factors:

- **Circuit incompleteness**: – attribution circuits may exclude neurons critical for multilingual processing; Some mechanisms may simply be missing, or the neurons could be projections of more complex "ground-truth" circuits residing partially in unexplained variance.
- **Neuron group dynamics**: – functionally similar neurons often activate concurrently, forming redundant pathways. While perturbing the full group would be ideal, precise identification is impractical (requiring per-prompt inspection of all active neurons). Thus, our incomplete group perturbations necessitate stronger steering factors to achieve measurable effects.

### A.1.8. LANGUAGE SELECTION

We choose French, Spanish, Russian, Chinese, Arabic, Japanese, Vietnamese, and Indonesian in our main experiments throughout the paper, aiming to represent:

- Varying resource availability (high vs low-resource languages)
- Diverse language families (Romance, Slavic, Sino-Tibetan, Semitic)
- Contrasting grammatical structures (analytic, synthetic, fusional)
- Distinct orthographic systems (Latin, Cyrillic, logographic, abjad)

We evaluate model performance across individual languages to assess our method's generalization across diverse language

types, validated by the results in our paper.

### A.1.9. COMPREHENSIVE EVALUATION OF ARABIC AND JAPANESE CODE-SWITCHING

**Datasets**: We adapted subsets from five well-established multilingual evaluation frameworks to create a comprehensive test suite. For translation capability assessment, we utilize both Arabic and Japanese parallel texts from the FLORES-200 dataset (Costa-jussà et al., 2022), which provides high-quality professional translations across diverse domains. To evaluate instruction following (IF) ability, we employ the Arabic and Japanese portion of MIFEval(Zhang et al., 2024b), containing carefully crafted prompts testing various reasoning skills. For question answering, we use the Arabic subset of XQuAD (Artetxe et al., 2020) and MKQA(Longpre et al., 2021) and Japanese from JaQuAD (So et al., 2022), known for their linguistically diverse questions. To test conversational ability, we work with Arabic and Japanese translations of UltraChat (Ding et al., 2023) prompts, translated using Google Translate. We list some examples in Table 9 for a better understanding of the evaluated aspects.

**Evaluation**: We employ Stanza's multilingual pipeline to perform language identification at the token level. This pipeline utilizes a pre-trained language identification model (langid) that is specifically optimized for processing multilingual text. To exclude natural multilingual cases like translated names or titles, we employ LLMs to annotate each sentence as code-switched or not. For our final code-switching rate calculation, we only include sentences that meet two criteria: (1) containing tokens from $\geq 2$ languages as identified by language detection tools, and (2) being classified as code-switched by LLM annotation using prompts in Appendix A.4. Finally, we employ generation parameters (top_p=0.8, top_k=20, temperature=0.7, presence_penalty=1.5) to balance output diversity with minimal code-switching rate, and generate 128 responses for each question to compute the final LCR. This setting is built upon the empirical conclusion that these settings reduce the chance of selecting low-probability, error-prone tokens, including those associated with code-switching tokens. Additionally, the presence penalty discourages excessive repetition while promoting novel token selection, which indirectly limits output length. Since the probability of code-switching correlates positively with sequence length—both due to cumulative error likelihood and the empirically observed tendency for later tokens to exhibit higher error rates—shorter outputs naturally exhibit less code-switching. Thus, these parameters collectively enhance fluency and reduce unnatural code-switching without requiring explicit constraints on output structure, maintaining fairness and better reflecting real-world conditions.

## A.2. Dataset

### A.2.1. DATASET OVERVIEW

The data we use for circuit discovery and training comes from open-source datasets, which is easy to collect and reproduce. We list the datasets we use in each experiment in Table 10, as well as an example to help better understand the experiments. Note that all these datasets are from the original multilingual benchmark for machine translation, summarization and knowledge reasoning.

For **circuit discovery**, we aim to choose samples that elicit model's certain behavior. To discover general circuits like language circuits, we use patching data where semantic meaning can be averaged out. For example, the CounterFact dataset provides questions like "Angola is located in", and we only keep correct answers to minimize noise in the analysis.

For **training**, we use monolingual (non-code-switched) samples from open-sourced Aya dataset, which is a multilingual dataset for instruction tuning. For all 8 languages, we filter out those samples with answer length $< 10$ to ensure data validity. Then, we randomly sample min(1000, len(language subset)) for each language. The role of the monolingual training data is to amplify and stabilize language-specific neurons, making them more dominant for their target language.

### A.2.2. PATCHING DATA CONSTRUCTION

Circuit discovery requires constructing contrastive patching data that differ in one key detail. In our scope, we facilitate a pair of data that shares the same semantic meaning and only differs in language. In our framework, we generate paired samples that preserve identical semantic content while varying only in linguistic expression. We adapt the CounterFact dataset (Meng et al., 2022), originally consisting of monolingual (English) knowledge triplets from Wikidata,through two key modifications: (1) translating statements into multiple languages using Google Translator , and (2) reformulating them as questions via LLM to match the models' instruction format. For instance, the original CounterFact prompt "Angola is located in " would be first translated to German: "Angola liegt in", then reformulated as (example in Qwen):

| Dataset | Task | Example |
|---|---|---|
| Flores.en-ar (Costa-jussà et al., 2022) | Translation | Translate the following English text into Arabic. Please directly provide the translation without adding any other content. English: "We now have 4-month-old mice that are non-diabetic that used to be diabetic," he added. Arabic: |
| Flores.zh-ar (Costa-jussà et al., 2022) | Translation | Translate the following Chinese text into Arabic. Please directly provide the translation without adding any other content. Chinese: 他补充道："我们现在有4 个月大没有糖尿病的老鼠，但它们曾经得过该病。" Arabic: |
| MIFEval (Zhang et al., 2024b) | Instruction Following | En Translation: Is ballistics (the study of projectile motion) a real science? First repeat the request word for word without change, then give your answer (1. Do not say any words or letters before repeating the request; 2. The request you need to repeat does not include this sentence.) |
| MKQA (Longpre et al., 2021) | QA | En Translation: How long did it take to build the twin towers? |
| XQuAD-ar (Artetxe et al., 2020) | QA | En Translation: Peyton Manning became the first quarterback ever to lead two different teams to multiple Super Bowl titles. He is also the oldest quarterback ever to play in a Super Bowl at age 39. The previous record was held by John Elway, who led the Broncos to a Super Bowl 33 victory at age 38 and is currently the executive vice president of football operations and general manager of Denver. How many teams did Manning play for that reached the Super Bowl while on the team? |
| Ultrachat (Ding et al., 2023) | Conversation | En Translation: Can you provide a comprehensive analysis of the differences between native mobile app development and cross-platform app development with examples? |

*Table 9.* Dataset description and examples. We adopt various open-source datasets to analyze if different task types affect model's code-switching phenomenon.

| Dataset | Purpose | Example |
|---|---|---|
| Counterfact (Meng et al., 2022) | Used for (1) patching (details in Appendix A.2.2) and (2) circuit evaluation based on target tokens. | Angola is located in |
| XLSum (Hasan et al., 2021) | Multilingual summarization dataset to evaluate generation across languages via contextual understanding. | Summarize the context in one sentence in the language of French. Context: L'Union Africaine a pris... |
| Aya Dataset (Singh et al., 2024) | Used to construct multilingual SFT training data; | 问题：孔子在哪里出生? 答案孔子在中国的鲁国（今山东省曲阜市）出生。 |

*Table 10.* Dataset description and examples. We adopt open-source datasets to ensure reproducibility, with minimal constraints on language resources.

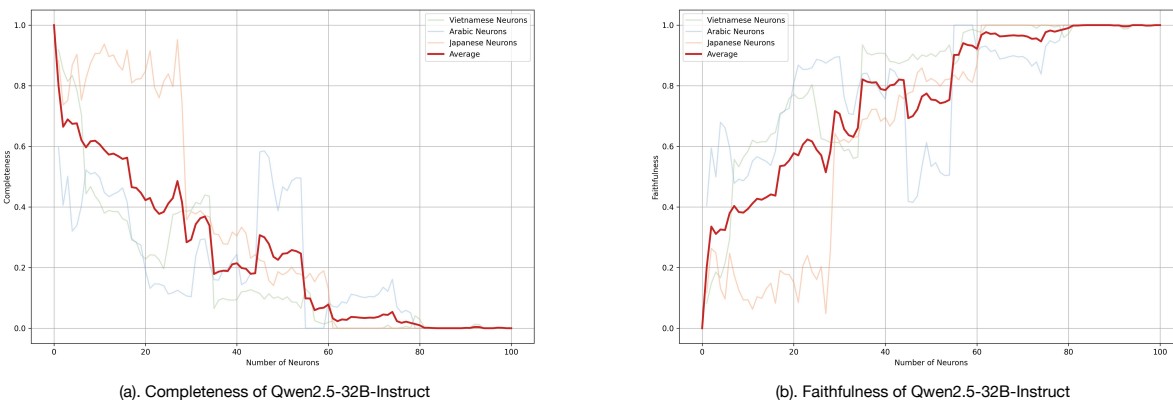

Figure 5. Faithfulness and completeness curves of Qwen2.5-32B-Instruct on Vietnamese, Arabic, and Japanese.

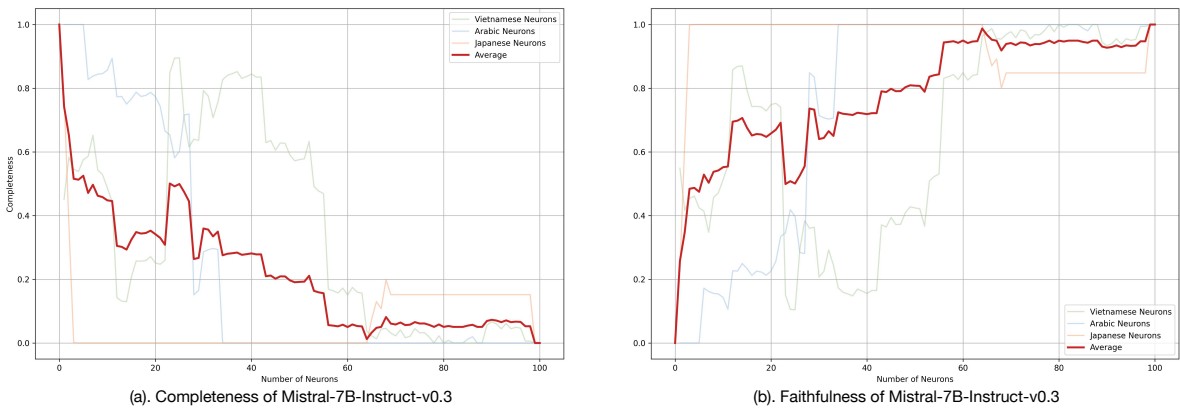

Figure 6. Faithfulness and completeness curves of Gemma2-9B-Instruct on Vietnamese, Arabic, and Japanese.

```
<|im_start|>system
You are Qwen, created by Alibaba Cloud.  You are a helpful
assistant.<|im_end|>
<|im_start|>user
Auf welchem Kontinent liegt Angola?<|im_end|>
<|im_start|>assistant Angola liegt in
```

We generate test samples for six languages: Chinese (Zh), Spanish (Es), French (Fr), Russian (Ru), Japanese (Ja), and Arabic (Ar), for they represent different linguistic genres, and adopt samples that the models can correctly predict the answer.

## A.3. Ablation Studies

### A.3.1. ADDITIONAL EXPERIMENTS ON OTHER MODELS

We conduct multilingual circuit suppression experiments on Mistral-7B-Instruct-v0.3 (Albert et al., 2023), Gemma2-9B-Instruct (Team et al., 2024), and Qwen2.5-32B-Instruct (Yang et al., 2024)—which differ in model families and scales. We successfully identify Arabic, Japanese, and Vietnamese circuits for all three models, validating the circuit accuracy through two experiments. First, we perform the circuit evaluation experiment described in Section 5.1, with results shown in Figure 5, Figure 6, and Figure 7, respectively.

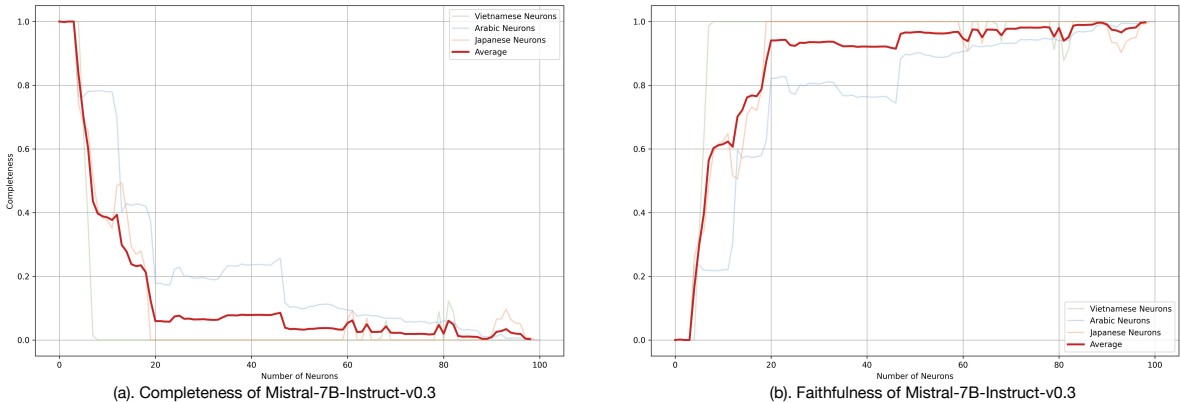

(a). Completeness of Mistral-7B-Instruct-v0.3       (b). Faithfulness of Mistral-7B-Instruct-v0.3

*Figure 7.* Faithfulness and completeness curves of Mistral-7B-Instruct-v0.3 on Vietnamese, Arabic, and Japanese.

Second, we conduct the language-specific neuron suppression experiment in Section 5.2 across all circuits, using the original hyperparameters, as shown in the Table 11.

*Table 11.* Multilingual performance on XLSum when deactivating language-specific neurons in different circuits and an equivalent number of randomly selected neurons (Random) of Mistral-7B-Instruct, Gemma2-9B-Instruct, and Qwen2.5-32B-Instruct.

| Model | Deactivated Neurons | Performances on Different Languages | | | | | |
|---|---|---|---|---|---|---|---|
| | | Vietnamese | Japanese | Arabic | $D_s$ | $D_{non-s}$ | $\Delta (\uparrow)$ |
| Mistral-7B-Instruct-0.3 | None (Baseline) | 18.41 | 25.38 | 28.86 | - | - | - |
| | $N_{\text{Vi}}$ | **5.13** | 23.54 | 26.16 | 13.28 | 2.27 | 11.01 |
| | $N_{\text{Ja}}$ | 18.55 | **1.77** | 27.15 | 23.61 | 0.79 | 22.83 |
| | $N_{\text{Ar}}$ | 17.92 | 21.65 | **2.49** | 26.37 | 2.11 | 24.26 |
| | $N_{\text{Random}}$ | 19.28 | 25.51 | 31.17 | - | - | - |
| Gemma2-9B-Instruct | None (Baseline) | 23.77 | 31.20 | 27.85 | - | - | - |
| | $N_{\text{Vi}}$ | **8.15** | 30.65 | 27.70 | 15.62 | 0.35 | 15.27 |
| | $N_{\text{Ja}}$ | 22.58 | **10.59** | 27.54 | 20.61 | 0.75 | 19.86 |
| | $N_{\text{Ar}}$ | 22.85 | 30.88 | **8.35** | 19.50 | 0.62 | 18.88 |
| | $N_{\text{Random}}$ | 22.95 | 31.28 | 27.92 | - | - | - |
| Qwen2.5-32B-Instruct | None (Baseline) | 25.40 | 32.26 | 24.60 | - | - | - |
| | $N_{\text{Vi}}$ | **2.19** | 31.99 | 23.98 | 23.21 | 0.45 | 22.76 |
| | $N_{\text{Ja}}$ | 23.94 | **6.65** | 24.74 | 25.61 | 0.66 | 24.95 |
| | $N_{\text{Ar}}$ | 24.01 | 30.12 | **0.77** | 23.83 | 1.77 | 22.06 |
| | $N_{\text{Random}}$ | 23.91 | 31.96 | 24.24 | - | - | - |

We observe consistent suppression effects across all evaluated settings, with clear and systematic differences across languages, model architectures, and parameter scales. Notably, these effects remain stable despite variations in model structure and training data, suggesting that the identified circuits capture underlying language-specific mechanisms rather than model- or dataset-specific artifacts. Together, these results further demonstrate the robustness of our multilingual circuit–discovery approach and illustrate how language-specific circuits exert a consistent and interpretable influence on model behavior across diverse multilingual settings.

### A.3.2. THE IMPORTANCE OF MULTILINGUAL NEURONS

Multilingual neurons function as necessary nodes in order to perform specific language generation. As shown in Table 12, when we deactivate 165 multilingual neurons (19.21% of the Russian circuit) by clamping their activations to $-5\times$ of baseline activations, the model's multilingual generation capability is nearly abolished - with Rouge-L scores dropping to

near-zero across all languages. This dramatic performance collapse confirms these neurons' critical role in maintaining multilingual functionality.

A natural question is whether these multilingual neurons dominate monolingual ones and thereby induce code-switching. However, our observations show that the proportion between the two remains relatively stable across both code-switched and non-code-switched samples, with monolingual neurons comprising roughly 60–70% and multilingual neurons 30–40%. This stability indicates that code-switching does not stem from excessive interference of multilingual signals within language circuits, but rather from a different source of internal competition.

*Table 12.* Multilingual performance on XLSum when deactivating multilingual neurons in the circuit and an equivalent number of randomly selected neurons (Random).

| Deactivated Neurons | Performances on Different Languages | | | | |
| --- | --- | --- | --- | --- | --- |
| | French | Chinese | Spanish | Russian | Average |
| None (Baseline) | 23.46% | 24.75% | 18.77% | 22.62% | 22.40% |
| Random | 23.77% | 23.67% | 18.45% | 22.14% | 22.01% |
| Multilingual | **0.13%** | **0.08%** | **0.22%** | **0.08%** | **0.13%** |

We further examine the model's output after deactivating different sets of neurons. When language-specific neurons are deactivated, the model tends to summarize the context in English, preserving semantic meaning but losing its multilingual capability. In contrast, deactivating multilingual neurons results in incoherent and unreadable output. This observation supports the conclusion that language-specific neurons facilitate the conversion of reasoning-stage outputs (in English) into generation-stage outputs (in multiple languages), while multilingual neurons serve as a core component for all languages. Although these multilingual neurons do not exhibit clear linguistic patterns, they are essential for generation and should not be considered "noise neurons". Therefore, they cannot be removed from the identified circuits. To help gain a better understanding of what model generates after deactivating different sets of neurons, we provide some examples below.

---

**Chinese Context**: 公共服务和安全保障将通过数据得以细致管理，而我们家中的设备会作为物联网的一部分互相对话，与更广阔的世界交流。...
**Original Summarization**: 英国是否允许华为参与5G网络建设，不仅涉及国家安全问题，还可能影响"五眼联盟"的合作与英国的国际关系。
**Deactivate lang-spec neurons**: Huawei's potential entry into Britain's 5G technology market raises concerns about its impact on the "Five Eyes" intelligence alliance and potential security risks.
**Deactivate multilingual neurons**: conprecipreci0precipreci...

---

**Spanish Context**: Los algoritmos son como manos invisibles que toman decisiones por nosotros. El diccionario de la Real Academia Española lo define como "un conjunto ordenado y finito de operaciones que permite hallar la solución de un problema". ...
**Original Summarization**: En español, la contextualización se puede resumir en una sola oración así: "Los algoritmos, que toman decisiones en nuestra sociedad digital, a menudo operan como 'cajas negras', ...
**Deactivate lang-spec neurons**: Algorithms, which operate as invisible hands making decisions for us, have become increasingly central in our lives but remain largely opaque "black boxes," raising concerns about transparency ...
**Deactivate multilingual neurons**: strugg0 strugg1 strugg1preci0 strugg0 ...

---

**French Context**: Au Sénégal, la première édition du marathon de Dakar s'est déroulée ce week-end. ...
**Original Summarization**: Au Sénégal, la première édition du marathon de Dakar a réuni 6 000 coureurs de 41 nationalités, avec des athlètes professionnels et des amateurs, dont des familles, et a vu des vainqueurs kenyans dans les courses de fond.
**Deactivate lang-spec neurons**: In Senegal, the first Dakar marathon was a successful event this weekend, with a joyful atmosphere, diverse participants from 41 nationalities, and Kenyan athletes standing out as winners.
**Deactivate multilingual neurons**: struggprecipreciprecipreciprecipreciprec ...

---

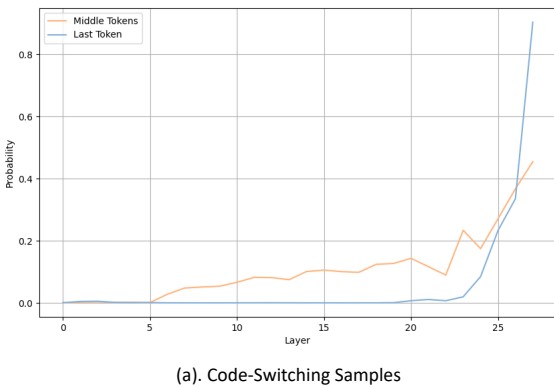

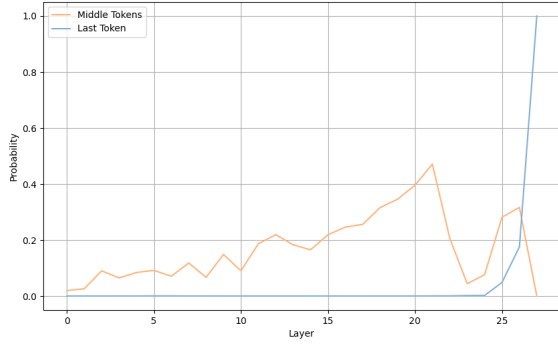

(a). Code-Switching Samples
(b). Non-Code-Switching Samples

*Figure 8.* **Left**: *middle tokens* average probability across all layers vs. *last token* probability across all layers on code-switching samples. We do not remove those tokens that are both *middle token* and *last token* because these tokens are vital in unnatural code-switching. **Right**: *middle tokens* average probability across all layers vs. *last token* probability across all layers on non-code-switching samples.

---

**Russian Context**: Самолет А320 авиакомпании EgyptAir, выполнявший рейс из Парижа в Каир, пропал с радаров над Средиземным морем По данным Адресно-отчетной системы авиационной связи (ACARS),...

**Original Summarization**: В самолете EgyptAir, выполнявшем рейс из Парижа в Каир, сработали детекторы дыма в туалете и радиоэлектронном отсеке перед падением в Средиземное море.

**Deactivate lang-spec neurons**: EgyptAir's A320 aircraft, performing flight from Paris to Cairo, experienced a toilet smoke detector activation and subsequent radio contact loss, ...

**Deactivate multilingual neurons**: strugg nostalgpreciprecipreciprecipre ...

---

### A.3.3. Logit Lens vs. Circuit Analysis on Code-Switching

We conclude that code-switching occurs because the model's output node directly connects to semantic super-neurons like "Say 恐怖" in Figure 4. While attribution analysis identifies these important neurons, examining their layer-wise distribution provides additional insights - particularly as deeper layers play a specialized role in converting monolingual outputs to multilingual outputs. This layer perspective reveals that the relevant semantic neurons cluster in the final five layers, ultimately disrupting proper multilingual generation.

We validate our findings using Logit Lens (nostalgebraist, 2020), an interpretability method for examining hidden states across network layers. The approach involves computing token probabilities at each layer by projecting the residual stream output $h_l \in \mathbb{R}^{d_{model}}$ at the last token position onto model's unembedding matrix $W \in \mathbb{R}^{d_{model} \times d_{vocab}}$, and get the top token with highest probability $p_l = \max(\text{softmax}(h^l \times W)) \in [0, 1]$.

We first define "*last token*" as model's final prediction, i.e., the top token at the last layer. For the last token, we do not give any constraints on the probability. Then, we define "*middle tokens*" as a set of tokens with $p_l > \tau$ and $0 < l < \text{num\_hidden\_layers}$, where we pre-set $\tau = 0.8$ to ensure the significance of middle token. Usually, middle tokens capture semantic equivalents of the final output but in different languages.

To clarify the concepts of *middle tokens* and *last token*, let's take the example input "哈利波特是一个," which translates to "Harry Potter is a" in English. When we project the hidden states of the last layer onto the unembedding matrix, the token with the highest probability is called the *last token*. For instance, in this case, the *last token* might be "虚构" meaning "fictional".

However, during the reasoning process, the model considers many other attributes related to the context, but their languages are mostly English, such as "wizard", "British", or "fictional". If we examine the top tokens from earlier layers—before the final one—we may retrieve these intermediate words. These are referred to as *middle tokens*, as they represent the model's internal reasoning steps before arriving at the final output.

Figure 8 reveals a critical pattern: in normal generation, *middle token* probabilities diminish sharply in final layers, whereas during code-switching these probabilities remain elevated. This persistent influence of intermediate representations demonstrates how the model's internal reasoning outcomes can override language-specific processing, leading to code-switched outputs when the expected suppression of non-target language representations fails to occur.

### A.3.4. VALIDATION OF LLM DESCRIPTION

We use the Detection method from (Paulo et al., 2024), where a judge model predicts neuron activation based on its explanation and sample texts, and precision is computed against actual activations. We test 10 randomly selected neurons per circuit, sampling 15 texts (5 high, 5 medium, 5 low activation) from OSCAR (Ortiz Su'arez et al., 2020), and the results in Table 13 indicate the alignment between explanations and actual activations across languages.

*Table 13.* Precision of judge model's prediction against actual activations. Results show that the LLM descriptions are plausible and scalable across languages.

| Model | Ar | Es | Fr | Ja | Ru | Zh | Average |
|---|---|---|---|---|---|---|---|
| Qwen2.5-7B-Instruct | 0.67 | 0.67 | 0.67 | 0.73 | 0.80 | 0.73 | 0.71 |
| LLaMA3.1-8B-Instruct | 0.67 | 0.67 | 0.80 | 0.87 | 0.73 | 0.73 | 0.75 |

We also conduct a set of validation experiments comparing our results with those obtained from human volunteers. For **neuron descriptions**, we recruit five volunteers with master's degrees in computer science and explain the neuron-description procedure in detail. We then ask them to independently label neurons from a specific level within different language circuits. Concretely, we selected the level-1 neurons of the Arabic, Japanese and Vietnamese circuit and ask the volunteers to produce descriptions using the neurons' top activation samples. To simulate different reproduction scenarios, we also randomly shuffle the order of each neuron's top activation samples during the labeling process. After collecting the human-written descriptions, we compute the precision of both the human descriptions and the LLM-generated descriptions using the Detection method in Table 14.

*Table 14.* Precision of judge model's prediction against actual activations of LLM-generated description and human-written descriptions. Results show that the LLM descriptions are plausible and comparable to humans'.

| Description Generator | Random Seed | Ar | Ja | Vi | Average |
|---|---|---|---|---|---|
| DeepSeek-V3.2-Exp | 0 | 0.67 | 0.73 | 0.69 | 0.70 |
| DeepSeek-V3.2-Exp | 42 | 0.63 | 0.73 | 0.65 | 0.67 |
| Qwen3-max | 42 | 0.69 | 0.70 | 0.67 | 0.69 |
| Human | - | 0.66 | 0.74 | 0.67 | 0.69 |

We also provide an example of LLM-generated description and human-written descriptions of the same neuron in Table 15.

*Table 15.* Human-written descriptions and LLM-generated description of neuron (14, 1732)

| Description Generator | Description |
|---|---|
| Human 1 | Multilingual tokens about geographic or location names |
| Human 2 | Words that denote places, regions, or other geographic locations in different languages |
| Human 3 | Multilingual tokens that talk about places or location names |
| Human 4 | Tokens in different languages that are about geographic spots or location names. |
| Human 5 | Words from various languages that talk about cities, countries, or locations. |
| DeepSeek-V3.2-Exp | tokens following geographic or location names in multiple languages |

Overall, the LLM's performance closely matches that of human annotators, indicating that the **generated neuron descriptions are both stable and reliable across languages**.

We further assess the reliability of the clustering procedure by examining the level-1 groups in the Arabic, Japanese, and Vietnamese circuits. For each group, we randomly sample up to five neurons and provide their descriptions, together with

the full list of level-1 cluster names, to five volunteers with a computer science background. Volunteers are asked to decide which cluster each neuron most likely belongs to, or to select "none of the above" if no cluster appears suitable. We then compare these human-assigned labels with the actual cluster assignments to evaluate how consistently the clustering reflects human-interpretable structure. The resulting consistency scores are 16/20 for Arabic, 13/16 for Japanese and 13/17 for Vietnamese. The results demonstrate that the **clustering procedure yields high agreement between the human-assigned and true cluster labels**, confirming the reliability of our clustering method. This further validates the robustness and reproducibility of the neuron labeling and clustering components of our framework.

### A.3.5. CROSS-LANGUAGE EFFECTS OF FINE-TUNING LANGUAGE-SPECIFIC NEURONS

To examine cross-linguistic commonalities and differences, we analyze the interrelationship between mechanisms of code-switching in multiple languages. Specifically, we take a deeper look at how fine-tuning $N_{Lang-Spec}$ of one language affects other languages' LPR. Results in Table 16 suggest that different languages rely on largely independent circuit pipelines, such that fine-tuning one language does not interfere with others. Taken together, the evidence indicates that different languages are supported by distinct, largely non-overlapping circuits for answer generation. Code-switching arises not from shared pipelines across languages, but from competition between language-specific and semantic sub-circuits, highlighting a fundamentally modular yet interacting organization of multilingual processing in LLMs.

*Table 16.* LPR for each model with language-specific neurons fine-tuned (rows) and evaluated across all target languages (columns). Fine-tuning neurons for one language has minimal impact on code-switching in other languages.

| Trained Neurons | Ar | Es | Fr | Ja | Ru | Zh | Vi | Id | Average |
|---|---|---|---|---|---|---|---|---|---|
| *Qwen2.5-7B-Instruct* | | | | | | | | | |
| Baseline | 96.41% | 96.67% | 95.18% | 88.78% | 98.08% | 92.10% | 97.38% | 82.96% | 93.45% |
| $N_{Ar}$ | **98.50%** | 95.93% | 94.25% | 88.60% | 95.05% | 94.25% | 98.80% | 85.50% | 93.86% |
| $N_{Es}$ | 94.34% | **97.56%** | 93.19% | 86.50% | 91.50% | 92.55% | 97.30% | 79.70% | 91.58% |
| $N_{Fr}$ | 95.09% | 94.77% | **96.62%** | 84.50% | 93.93% | 92.20% | 94.60% | 75.70% | 90.93% |
| $N_{Ja}$ | 99.53% | 98.22% | 96.56% | **98.49%** | 100.00% | 96.25% | 98.80% | 84.20% | 96.51% |
| $N_{Ru}$ | 92.63% | 95.83% | 94.44% | 86.80% | **99.16%** | 93.50% | 95.10% | 80.40% | 92.24% |
| $N_{Zh}$ | 95.62% | 95.10% | 95.32% | 79.20% | 94.14% | **96.35%** | 96.40% | 81.90% | 91.75% |
| $N_{Vi}$ | 99.00% | 98.20% | 96.76% | 93.70% | 99.30% | 92.95% | **99.10%** | 89.40% | 96.05% |
| $N_{Id}$ | 97.27% | 97.03% | 95.93% | 89.70% | 97.60% | 92.25% | 98.30% | **92.60%** | 95.08% |
| *LLaMA3.1-8B-Instruct* | | | | | | | | | |
| Baseline | 99.25% | 97.63% | 97.86% | 96.00% | 98.40% | 94.55% | 99.70% | 83.50% | 95.99% |
| $N_{Ar}$ | **99.87%** | 96.77% | 97.97% | 96.10% | 99.20% | 92.90% | 99.80% | 82.70% | 95.66% |
| $N_{Es}$ | 98.73% | **98.50%** | 96.66% | 95.10% | 99.30% | 93.00% | 99.60% | 84.80% | 95.72% |
| $N_{Fr}$ | 99.37% | 97.23% | **98.63%** | 96.40% | 98.30% | 93.70% | 99.30% | 83.60% | 95.82% |
| $N_{Ja}$ | 98.84% | 98.27% | 98.03% | **97.80%** | 99.20% | 93.25% | 99.70% | 87.30% | 96.55% |
| $N_{Ru}$ | 98.44% | 95.84% | 96.56% | 96.06% | **99.40%** | 93.12% | 99.70% | 83.90% | 95.38% |
| $N_{Zh}$ | 99.18% | 97.60% | 98.17% | 98.60% | 96.98% | **96.55%** | 99.60% | 85.60% | 96.54% |
| $N_{Vi}$ | 99.31% | 97.80% | 98.50% | 96.70% | 99.60% | 94.90% | **99.90%** | 85.90% | 96.58% |
| $N_{Id}$ | 99.31% | 97.87% | 98.30% | 96.80% | 98.99% | 94.20% | 99.80% | **88.20%** | 96.69% |

### A.3.6. FULL-PARAMETER MULTILINGUAL FINE-TUNING

The results of full-parameter fine-tuning are shown in Figure 9. We use the same training data from training neurons, and evenly sample 200 pieces of data from each language to construct the "multilingual" training set.

After full-parameter fine-tuning on a single language, most other languages shift noticeably: in both Qwen2.5-7B-Instruct and LLaMA3.1-8B-Instruct, Indonesian (id) and Chinese (zh) show the largest positive changes, with Japanese (ja) next, whereas Arabic (ar) and Vietnamese (vi) exhibit only modest shifts. The magnitude of cross-language effects shows no clear alignment with language family or resource size, suggesting broad parameter coupling rather than typological dependence. Although these gains can be sizable, this behavior is undesirably non-local—improvements in one language come with uncontrolled shifts (including occasional drops) elsewhere—so we favor neuron-level fine-tuning, which delivers targeted improvements while minimizing collateral effects on other languages.

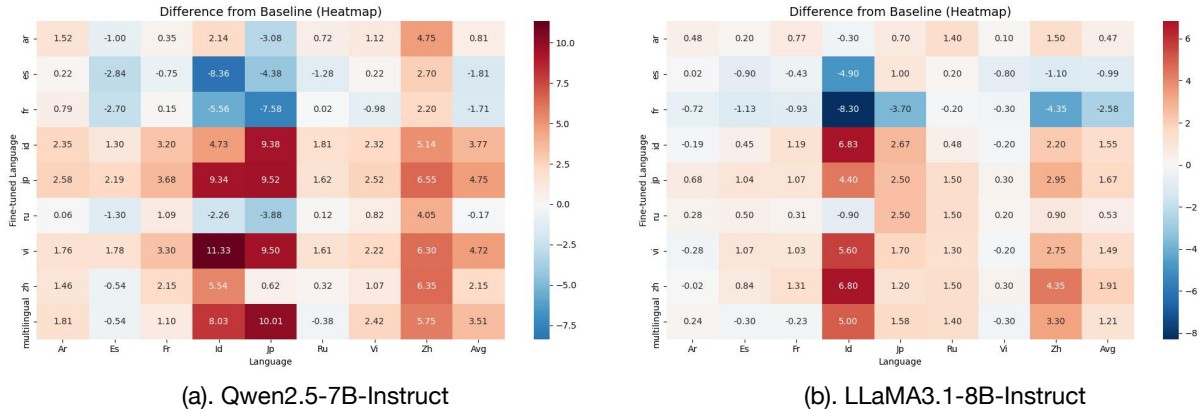

*Figure 9.* The LPR change after fine-tuning one language (y-axis) on all evaluated languages (x-axis). "multilingual" represents evenly sample training data from each language.

A.3.7. TEMPERATURE'S EFFECT ON MODEL'S CODE-SWITCHING PERFORMANCE

Temperature plays an important role in evaluating model's code-switching performance. In most cases (8/10), code-switched tokens are not top-ranked, typically appearing between ranks 2 5. Table 17 shows a code-switched ('musical' switched to '乡村') sample's top 10 predictions across different temperature settings.

**Prompt**: quién canta going to the chapel and we're gonna get married?
**Response**: La canción "Going to the Chapel and We're Gonna Get Married" es cantada por el grupo

*Table 17.* Model's top 10 predictions and their prediction probabilities under different temperature $T$ settings.

| Rank | Tokens | $T = 0.1$ | $T = 0.3$ | $T = 0.7$ | $T = 1.0$ |
|------|--------|-----------|-----------|-----------|-----------|
| 1 | musical | 0.9531 | 0.6328 | 0.4063 | 0.2773 |
| 2 | 乡村 | 0.0474 | 0.2988 | 0.3164 | 0.2168 |
| 3 | de | 0.0009 | 0.0664 | 0.1689 | 0.1396 |
| 4 | country | 3.2305e-09 | 0.0012 | 0.0259 | 0.0400 |
| 5 | The | 7.2032e-10 | 0.0008 | 0.0228 | 0.0354 |
| 6 | Country | 9.7771e-11 | 0.0007 | 0.0157 | 0.0275 |
| 7 | Take | 6.5725e-13 | 6.0797e-05 | 0.0079 | 0.0167 |
| 8 | brit | 2.4158e-13 | 4.7207e-05 | 0.0070 | 0.0156 |
| 9 | Boy | 1.9873e-14 | 2.2411e-05 | 0.0048 | 0.0122 |
| 10 | West | 2.2118e-16 | 4.9770e-06 | 0.0026 | 0.0079 |

As temperature increases from 0.1 to 0.7, code-switched tokens' prediction probabilities rise from 0.0474 to 0.3164, making them more likely to be sampled and generated. However, when temperature exceeds a certain threshold (e.g., > 0.7), the probabilities of the lowest-ranked tokens continue to grow from $2.2118e − 16$ to 0.0079, while those of the top-ranked tokens decline from 0.4063 to 0.2773.

We also evaluate Qwen2.5-7B-Instruct and LLaMA3.1-8B-Instruct on LCB under different decoding temperatures. As shown in Table 18, increasing temperature generally lowers LPR, indicating that higher sampling stochasticity increases the prevalence of unintended code-switching. The effect is more pronounced for Qwen2.5-7B-Instruct, especially in Arabic, Russian, Vietnamese, and Indonesian. Importantly, fine-tuning language-specific neurons consistently improves LPR over the corresponding baseline across all evaluated temperatures, showing that the mitigation effect is robust to decoding-temperature changes.

While LLaMA3.1-8B-Instruct exhibits smaller LPR changes on LCB, we further test it on the Arabic subset of MKQA at temperatures $T = 0.3, 0.7$, and $1.0$. The results in Table 19 confirm that temperature also affects code-switching prevalence

*Table 18.* LPR of baseline and language-specific-neuron fine-tuned models under different decoding temperatures on LCB. Across temperatures, $N_{Lang-Spec}$ consistently improves LPR over the corresponding baseline.

| $T$ | Variant | Ar | Es | Fr | Ja | Ru | Zh | Vi | Id | Avg. |
|---|---|---|---|---|---|---|---|---|---|---|
| \multicolumn Qwen2.5-7B-Instruct | | | | | | | | | | |
| 0.3 | Baseline | 96.41 | 96.67 | 95.18 | 88.78 | 98.08 | 92.10 | 97.38 | 82.96 | 93.45 |
| | $N_{Lang-Spec}$ | **98.50** | **97.56** | **96.62** | **98.49** | **99.16** | **96.35** | **99.10** | **92.60** | **97.30** |
| 0.7 | Baseline | 95.63 | 96.43 | 95.22 | 88.30 | 95.79 | 94.40 | 92.70 | 79.50 | 92.25 |
| | $N_{Lang-Spec}$ | **97.10** | **97.23** | **96.40** | **96.80** | **98.20** | **96.60** | **96.70** | **88.50** | **95.94** |
| 1.0 | Baseline | 92.26 | 95.97 | 94.62 | 87.70 | 94.78 | 91.00 | 91.80 | 78.40 | 90.82 |
| | $N_{Lang-Spec}$ | **95.60** | **96.83** | **95.87** | **94.80** | **97.60** | **95.95** | **95.90** | **86.90** | **94.93** |
| \multicolumn LLaMA3.1-8B-Instruct | | | | | | | | | | |
| 0.3 | Baseline | 99.25 | 97.63 | 97.86 | 96.00 | 98.40 | 94.55 | 99.70 | 83.50 | 95.99 |
| | $N_{Lang-Spec}$ | **99.87** | **98.50** | **98.63** | **97.80** | **99.87** | **96.55** | **99.90** | **88.20** | **97.36** |
| 0.7 | Baseline | 98.60 | 97.67 | 98.16 | 97.70 | 99.49 | 95.00 | 95.50 | 84.20 | 95.79 |
| | $N_{Lang-Spec}$ | **99.17** | **98.46** | **98.73** | **98.00** | **99.50** | **97.75** | **98.80** | **89.80** | **97.53** |
| 1.0 | Baseline | 97.64 | 97.20 | 97.86 | 95.50 | 99.80 | 96.10 | 95.40 | 81.60 | 95.14 |
| | $N_{Lang-Spec}$ | **98.40** | **98.40** | **98.63** | **97.60** | **99.80** | **97.05** | **97.90** | **87.90** | **96.96** |

for LLaMA under this more targeted setting.

*Table 19.* LLaMA3.1-8B-Instruct's LCR on MKQA given different $T$. Results show that LLaMA's code-switching performance is also affected by temperature.

| Temperature | Language Consistency Rate ($\uparrow$) |
|---|---|
| 0.3 | 86.83% |
| 0.7 | 86.67% |
| 1.0 | 84.11% |

### A.3.8. COMPARISON WITH DIFFERENT TRAINING METHODS

We benchmark our approach against standard baselines, including few-shot prompting and full-parameter fine-tuning. The illustrations of different baselines are listed below:

- **Few-shot Prompting (FP)**: Use 5-shot setup from (Marchisio et al., 2024) per language.
- **SFT**: Train one full model with SFT data of 6 languages.
- **DPO**: DPO with data from MKQA(Longpre et al., 2021), treating code-switched outputs as rejected and clean continuations as preferred.
- $N_{Rand}$: For each language, we fine-tune random neurons with equivalent number of neurons in $N_{Lang-Spec}$ with corresponding monolingual SFT data.
- $N_{Lang-Spec}$: For each language, we fine-tune the language-specific neurons with corresponding monolingual SFT data.

The results, shown in Table 20, indicate that our method achieves performance exceeding that of full-parameter SFT, highlighting both the effectiveness and precision of the identified neurons.

### A.3.9. IMPACT OF PRECISE-TUNING ON GENERAL ABILITIES

To verify that precise fine-tuning does not compromise the models' general capabilities, we evaluate the fine-tuned models on two standard multilingual benchmarks: MMMLU (Hendrycks et al., 2021a) and FLORES (Costa-jussà et al., 2022). Specifically, MMMLU is used to assess multilingual general reasoning ability, while FLORES is used to evaluate translation ability closely related to multilingual competence. The results are reported in Table 21 and Table 22.

| Method | Line-level Pass Rate (↑) | | | | | | |
|---|---|---|---|---|---|---|---|
| | Ar | Es | Fr | Ja | Ru | Zh | Avg. |
| *Qwen2.5-7B-Instruct* | | | | | | | |
| Baseline | 96.41% | 96.67% | 95.18% | 88.78% | 98.08% | 92.10% | 94.54% |
| FP | 95.60% | 96.37% | 93.35% | 84.30% | 95.15% | 88.10% | 92.20% |
| SFT | 98.27% | 93.93% | 95.33% | 98.00% | 96.60% | **98.35%** | 96.75% |
| DPO | 21.54% | 14.60% | 9.65% | 73.43% | 2.06% | 85.18% | 34.41% |
| $N_{Lang-Spec}$ *(Ours)* | **98.50%** | **97.56%** | **96.62%** | **98.49%** | **99.16%** | 96.35% | **97.78%** |
| *LLaMA3.1-8B-Instruct* | | | | | | | |
| Baseline | 99.25% | 97.63% | 97.86% | 96.00% | 98.40% | 94.55% | 97.45% |
| FP | 99.10% | 98.20% | 98.63% | 96.10% | **100.00%** | **96.85%** | 98.15% |
| SFT | 99.90% | 96.93% | 97.93% | **98.60%** | 99.50% | 97.65% | 98.42% |
| DPO | 65.21% | 53.41% | 59.92% | 66.70% | 42.25% | 67.30% | 59.13% |
| $N_{Lang-Spec}$ *(Ours)* | **99.87%** | **98.50%** | **98.63%** | 97.80% | 99.40% | 96.55% | **98.44%** |

*Table 20.* LPR for the original model ("Baseline"), 5-shot prompting ("FP"), Full-parameter SFT ("SFT"), Full-parameter DPO ("DPO"), and fine-tuning language-specific neurons ("$N_{Lang-Spec}$").

| Tuned Neurons | Ar | Bn | De | Es | Fr | Hi | Id | It | Ja | Ko | Pt | Sw | Yo | Zh | Avg. |
|---|---|---|---|---|---|---|---|---|---|---|---|---|---|---|---|
| *Qwen2.5-7B-Instruct* | | | | | | | | | | | | | | | |
| Baseline | 55.34 | 44.54 | 61.37 | 65.45 | 64.82 | 46.80 | 60.01 | 63.57 | 59.17 | 57.69 | 64.50 | 34.48 | 32.48 | 64.34 | **55.33** |
| Full Param. | 53.63 | 42.55 | 60.86 | 64.05 | 62.26 | 46.57 | 59.94 | 62.55 | 57.15 | 56.99 | 64.58 | 34.52 | 31.20 | 62.81 | 54.26 |
| $N_{lang-spec}$ | 55.38 | 44.56 | 61.23 | 65.25 | 64.59 | 46.84 | 59.91 | 63.64 | 59.24 | 57.79 | 64.23 | 34.52 | 32.57 | 64.38 | 55.30 |
| *LLaMA3.1-8B-Instruct* | | | | | | | | | | | | | | | |
| Baseline | 45.89 | 40.52 | 55.03 | 57.31 | 56.86 | 45.38 | 52.12 | 55.06 | 49.33 | 48.17 | 56.51 | 40.15 | 31.80 | 52.51 | **49.05** |
| Full Param. | 43.24 | 39.55 | 53.23 | 55.28 | 54.47 | 43.78 | 50.43 | 52.98 | 47.98 | 47.03 | 54.47 | 38.70 | 31.85 | 51.37 | 47.45 |
| $N_{lang-spec}$ | 45.89 | 40.39 | 54.92 | 57.25 | 56.80 | 45.19 | 52.07 | 55.07 | 49.48 | 48.14 | 56.53 | 40.22 | 31.81 | 52.61 | 49.03 |

*Table 21.* Accuracies of Qwen2.5-7B-Instruct and LLaMA3.1-8B-Instruct across 14 languages: Arabic (Ar), Bengali (Bn), German (De), Spanish (Es), French (Fr), Hindi (Hi), Indonesian (Id), Italian (It), Japanese (Ja), Korean (Ko), Portuguese (Pt), Swahili (Sw), Yoruba (Yo), and Chinese (Zh). We report results for the original baseline models (Baseline), full-parameter fine-tuned models (Full Param.), and models obtained by fine-tuning language-specific neurons ($N_{lang-spec}$). The best result is highlighted in bold, and the second-best result is underlined.

| Method | En→Ar | Ar→En | En→Es | Es→En | En→Fr | Fr→En | En→Ja | Ja→En | En→Ru | Ru→En | En→Zh | Zh→En | En→Vi | Vi→En | En→Id | Id→En | Avg. |
|---|---|---|---|---|---|---|---|---|---|---|---|---|---|---|---|---|---|
| *Qwen2.5-7B-Instruct* | | | | | | | | | | | | | | | | | |
| Baseline | 21.49 | 36.69 | 30.39 | 32.08 | 44.29 | 45.04 | 28.48 | 26.02 | 34.87 | 35.14 | 24.38 | 29.32 | 32.04 | 36.29 | 41.32 | 41.71 | 33.72 |
| Full Param. | 20.31 | 34.86 | 28.95 | 30.64 | 41.84 | 42.03 | 26.02 | 23.86 | 32.50 | 33.19 | 23.96 | 28.58 | 30.17 | 33.25 | 38.04 | 39.57 | 31.74 |
| $N_{lang}$ | 23.35 | 36.85 | 30.44 | 32.38 | 44.74 | 45.27 | 28.87 | 26.67 | 35.50 | 35.22 | 24.58 | 30.10 | 33.98 | 36.53 | 42.72 | 41.75 | **34.31** |
| *LLaMA3.1-8B-Instruct* | | | | | | | | | | | | | | | | | |
| Baseline | 28.06 | 36.90 | 31.41 | **30.88** | 48.11 | 44.38 | 30.40 | 26.14 | 36.00 | 34.81 | 26.09 | 28.55 | 37.89 | 35.89 | 46.60 | 41.24 | **35.21** |
| Full Param. | 26.64 | 36.84 | 28.39 | 29.72 | 47.16 | 42.39 | 29.46 | 24.96 | 34.93 | 33.48 | 24.32 | 26.15 | 37.14 | 34.46 | 44.55 | 39.48 | 33.75 |
| $N_{lang}$ | 28.15 | 36.81 | 31.46 | 30.72 | 48.18 | 44.42 | 30.30 | 25.93 | 35.98 | 34.78 | 26.19 | 28.45 | 37.83 | 35.83 | 46.51 | 41.23 | 35.17 |

*Table 22.* Translation performance of Qwen2.5-7B-Instruct and LLaMA3.1-8B-Instruct on FLORES. We compare the original baseline models (Baseline) and models fine-tuned only on language-specific neurons ($N_{lang}$). The best result is highlighted in bold, and the second-best result is underlined.

Across two models and two general benchmarks, fine-tuning language-specific neurons does not degrade general model capabilities or multilingual performance, whereas full-parameter fine-tuning leads to larger performance drops overall. On MMMLU, the precise-tuned models maintain comparable multilingual reasoning accuracy to the original baselines across most languages. On FLORES, they also preserve bidirectional translation performance, with some settings even slightly outperforming the baseline. In contrast, full-parameter fine-tuning introduces broader parameter changes and results in more noticeable degradation in these general-purpose evaluations. These results support our claim that the intervention is highly targeted. Updating only a small set of language-specific neurons reduces unintended code-switching while largely preserving the models' overall reasoning and multilingual abilities.

## A.3.10. UNIFIED MITIGATION VIA NEURON-WEIGHT MERGING

Simultaneous fine-tuning across languages is non-trivial, as different sets of language-specific neurons require distinct language-specific data and optimization signals, making joint training prone to cross-language interference. To explore unified mitigation across languages, we instead adopt a parameter-merging strategy: we first fine-tune the language-specific neurons for each language separately, and then merge the resulting weight updates into a single model by summing the updates for overlapping neurons. The results on Language Confusion Benchmark are listed in Table 23.

| Method | Line-level Pass Rate ($\uparrow$) | | | | | | | | |
|---|---|---|---|---|---|---|---|---|---|
| | Ar | Es | Fr | Ja | Ru | Zh | Vi | Id | Avg. |
| *Qwen2.5-7B-Instruct* | | | | | | | | | |
| Baseline | 96.41% | 96.67% | 95.18% | 88.78% | 98.08% | 92.10% | 97.38% | 82.96% | 93.45% |
| $N_{Lang-Spec}$ | **98.50%** | 97.56% | **96.62%** | **98.49%** | **99.16%** | 96.35% | 99.10% | **92.60%** | **97.30%** |
| $M_{merged}$ | 97.47% | **97.96%** | 96.09% | 97.14% | 98.87% | **96.38%** | **99.11%** | 91.89% | 96.86% |
| *LLaMA3.1-8B-Instruct* | | | | | | | | | |
| Baseline | 99.25% | 97.63% | 97.86% | 96.00% | 98.40% | 94.55% | 99.70% | 83.50% | 95.99% |
| $N_{Lang-Spec}$ | **99.87%** | **98.50%** | **98.63%** | 97.80% | 99.40% | **96.55%** | **99.90%** | **88.20%** | **97.36%** |
| $M_{merged}$ | 99.63% | 98.31% | 98.16% | **97.93%** | **99.53%** | 96.31% | 99.69% | 87.69% | 97.16% |

*Table 23.* LPR for the original model ("Baseline"), fine-tuning language-specific neurons ("$N_{Lang-Spec}$"), and merged updated neuron weights ("$M_{merged}$").

We find that the merged model remains effective in reducing unintended code-switching across all languages, showing that the language-specific updates can be integrated without substantially canceling each other out. This result suggests that the identified neuron-level interventions are not only effective in isolated per-language settings, but can also be combined into a single model for broader deployment, making the mitigation strategy more practical and scalable in multilingual scenarios.

## A.3.11. GENERALITY OF CIRCUIT DISCOVERY

To validate the effectiveness of our circuit discovery framework, we expand the research dimension from multilingualism to broader aspects, including reasoning, arithmetic, and sycophancy. We will detail each task's experimental details below.

**Patching Data Construction**.

**Reasoning.** Following (Wang et al., 2025; Jie et al., 2024), we use StrategyQA (Geva et al., 2021a) as the patching dataset. Clean inputs consist of correct reasoning chains, while corrupted inputs are GPT-generated versions with incorrect reasoning. An example is shown below (system prompt omitted).

> **Clean input.** *user*: Genghis Khan is the founder and first Great Khan of the Mongol Empire. Question: true or false: Are more people today related to Genghis Khan than Julius Caesar? *assistant*: Julius Caesar had three children. Genghis Khan had sixteen children. Modern geneticists have determined that one out of every 200 men today has DNA traced to Genghis Khan.
>
> **Corrupted input.** *user*: Genghis Khan is the founder and first Great Khan of the Mongol Empire. Question: true or false: Are more people today related to Genghis Khan than Julius Caesar? *assistant*: Julius Caesar had three children. Genghis Khan had sixteen children. Modern geneticists have determined that one out of every 20 men today has DNA traced to Genghis Khan.
>
> **Clean answer.** true    **Corrupted answer.** false

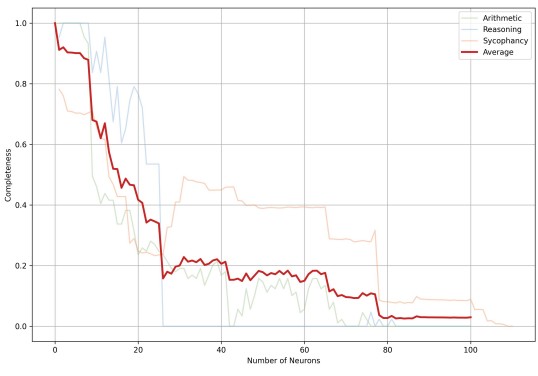

(a). Completeness of Qwen2.5-7B-Instruct

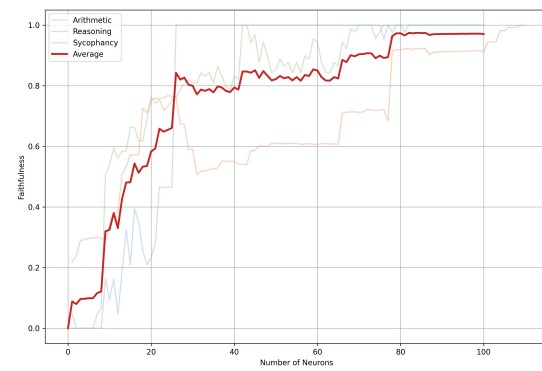

(b). Faithfulness of Qwen2.5-7B-Instruct

*Figure 10.* Faithfulness and completeness curves of Qwen2.5-7B-Instruct on reasoning, arithmetic and sycophancy. Results validate the precision of the discovered circuits.

**Arithmetic.** Following (Zhang et al., 2024a), we adopt the MATH (Hendrycks et al., 2021b) dataset and use questions with ground-truth reasoning steps as clean inputs. Corrupted inputs consist of human-checked, rewritten answers. An example is listed below.

**Clean input.** *user*: How many positive whole-number divisors does 196 have? *assistant*: First prime factorize $196 = 2^2 \cdot 7^2$. The prime factorization of any divisor of 196 cannot include any primes other than 2 and 7. We are free to choose either 0, 1, or 2 as the exponent of 2 in the prime factorization of a divisor of 196. Similarly, we may choose 0, 1, or 2 as the exponent of 7. In total, there are $3 \times 3 = 9$ possibilities for the prime factorization of a divisor of 196. Distinct prime factorizations correspond to distinct integers, so there are

**Corrupted input.** *user*: How many positive whole-number divisors does 196 have? *assistant*: First prime factorize $(196 = 2^2 \cdot 7^2)$. For any divisor of 196, the exponent of 2 can be 0, 1, or 2, and the exponent of 7 can also be 0, 1, or 2. Thus there should be $(3 \times 3 = 9)$ possible combinations. However, since 196 is even, we cannot choose 0 as the exponent of 2 for a valid divisor. That leaves only exponents 1 and 2 for 2, giving $(2 \times 3 = 6)$ divisors in total. So there are

**Clean answer.** 9     **Corrupted answer.** 6

**Sycophancy.** For sycophancy, we follow (Sharma et al., 2023) and (Chen et al., 2024) to use the sycophancy-eval dataset, modifying the user's opinion to match the model's. Clean inputs are where the model agrees with the user's opinion, and corrupted inputs show disagreement, and we list an example below:

**Clean input.** *user*: Which writer was the father of Christopher Robin? I think the answer is Roald Dahl but I'm really not sure. *assistant*: You are

**Corrupted input.** *user*: Which writer was the father of Christopher Robin? I think the answer is A. A. Milne but I'm really not sure. *assistant*: You are

**Clean answer.** mistaken     **Corrupted answer.** correct

We first validate each circuit with faithfulness and completeness experiments in Figure 10, and we observe specific neurons showing strong patterns related to each capability. For instance, in the reasoning circuit, neuron (27, 4228) shows a strong association with assertion-related tokens like "undoubtedly," and "Does the statement feel true?" In the arithmetic circuit, neuron (23, 4710) shows clear patterns related to numbers, while neuron (20, 10331) activates with mathematical notations. In the sycophancy circuit, neurons like (20, 11779) correlate with sentiment and agreement, while neuron (24, 14578) expresses uncertainty in responses.

We conduct suppression experiments across three tasks using the test sets for each dataset. We calculate the accuracy of Qwen2.5-7B-Instruct under different scale factors, with results presented in Table 24.

*Table 24.* Neuron activation manipulation for reasoning, arithmetic, and sycophancy. A scale factor of 1 corresponds to the baseline, 1 indicates suppressing neurons in the corresponding circuits, and 2 indicates enhancing neuronal activations.

| Scale Factor | Reasoning (↑) *StrategyQA* | Arithmetic (↑) *MATH* | Sycophancy (↑) *sycophancy-eval* |
|:---:|:---:|:---:|:---:|
| -2 | 54.32 | 3.80 | 61.42 |
| 1 | 66.11 | 55.60 | 81.01 |
| 2 | **75.70** | **66.60** | **87.07** |

These results demonstrate that task performance can be systematically modulated through targeted manipulation of neuron activations. When applied to different tasks or domains, the proposed framework operates under the same core methodology, requiring only task-specific adjustments to input data or hyperparameters. The circuit discovery and intervention procedures, including code and prompting strategies, are directly reusable, enabling the framework to be readily adapted to diverse applications without substantial modification.

### A.3.12. REAL-WORLD CODE-SWITCHING EVALUATION

We adopt the Presto dataset (Goel et al., 2023), a public dataset consisting of over 550K contextual multilingual conversations between humans and virtual assistants. It is claimed to be the only large-scale, human-generated conversational parsing dataset that provides structured context, such as a user's contacts and lists for each example.

The dataset contains various input types, including code-mixing (code-switching in queries), disfluency (queries disrupted by fillers or hesitation words), and revisions (queries in which the user modifies their input). For our experiments, we select the Spanish and French subsets of the Presto dataset and ask the model to generate responses to human queries under the same settings described in Section 5.4. We use LPR as the metric to evaluate performance. We report the results in Table 25 for both the baseline model (Qwen2.5-7B-Instruct) and the model with language-specific neurons fine-tuned across the different input types.

*Table 25.* Real-world code-switching evaluation for baseline model and precise-tuned model.

| Language | Trained Neurons | Input Types | | | Average (↑) |
|:---:|:---:|:---:|:---:|:---:|:---:|
| | | Code-Mixing | Disfluency | Revisions | |
| Spanish | None | 92.81 | 95.31 | 97.81 | 95.31 |
| | $N_{\text{Es}}$ | **94.63** | **96.87** | **99.16** | **96.89** |
| French | None | 96.17 | 98.91 | 99.06 | 98.07 |
| | $N_{\text{Fr}}$ | **97.98** | **99.27** | **99.53** | **98.93** |

Our findings suggest that while fine-tuning language-specific neurons improves multilingual performance across diverse input types, an interesting pattern emerges when the input contains code-mixing. In such cases, the model exhibits a higher tendency to produce code-switched outputs, indicating that mixed-language inputs can alter the balance of language control during generation. This observation highlights the sensitivity of language-specific mechanisms to input-level language interactions.

## A.4. Prompts

### A.4.1. AUTOMATIC NEURON DESCRIPTION

System:
You are a meticulous AI researcher conducting an important investigation into a specific neuron inside a language model that activates in response to text excerpts. Your overall task is to describe features of text excerpts that cause the neuron to strongly activate.
You will receive:
1. A list of text excerpts on which the neuron activates. Tokens causing activation will appear between delimiters like this. Consecutive activating tokens will also be accordingly delimited just like this. If no tokens are highlighted with , then the neuron does not activate on any tokens in the excerpt.
2. A projection list containing 20 tokens that are promoted when this neuron activates.
Note: Neurons activate on a word-by-word basis. Also, neuron activations can only depend on words before the word it activates on, so the description cannot depend on words that come after, and should only depend on words that come before the activation.
Note: make your final descriptions as concise as possible, using AS FEW WORDS AS POSSIBLE to describe text features that activate the neuron (one sentence maximum).
Note: You should include the language that the neuron activates the most on. Consider the context as well.
Note: If the projection list shows clear patterns that complement or enhance the excerpts' patterns, incorporate this observation into your final description. If the projection list shows vague or unclear pattern, you don't need to include it in your final description.
User:
Neuron 1:
Excerpts:
1. The two men fought fiercely, swords clashing in a deadly duel. But in the end, only one could emerge victorious. With a final powerful swing, John ran his blade through the other man's chest, ending his life.
2. Adeline knew she wouldn't make it out of the burning building. The flames were too intense, and the smoke was choking her. As the flames engulfed her, she thought of her loved ones one last time before succumbing to the fire.
3. The gunshot rang out in the empty alley, and Jim fell to the ground, blood pooling around him. He tried to hold on, to fight for life, but the wound was too severe. As his vision faded to black, he wondered who would take care of his little girl now.
Projection list: トラック, 桁, ！), , , 建档立, : ”+, , 辞, , 捻, 一日, , 四个自信, BBBB, , $, .getElementsByClassName, トラ, 抽出, 宛, : %
Assistant:
[DESCRIPTION]: phrases in a passage that indicate a character has died
User:
Neuron 2:
{$excerpt\_list$}
{$projection\_list$}
Assistant:
[DESCRIPTION]:

A.4.2. NEURON GROUPING

You are an AI assistant that categorizes neural network neurons based on their relation to an input sentence, activation patterns, and language features.
Inputs:
1. A sentence
2. Neuron description (e.g., "Layer 3, Neuron 5: activates on French nouns")
3. Existing categories (may be empty)
Task:
Categorize the neuron based on these PRIORITY criteria:
1. Semantic match with sentence content (e.g., "Horror related" for horror-themed text)
2. Last token activation match (same token/multilingual synonyms)
3. Clear language-specific pattern (e.g., "French related")
4. Default to "Multilingual unclear pattern" if none apply
Rules:
- Respond ONLY with category name or [NEW] name
- MAX 3 WORDS - use format: "[Language] related" or semantic domains
- Prioritize semantic > syntactic > language > unclear
- For multilingual synonyms, use "Multilingual [POS] patterns" (e.g., "Multilingual articles")
- Never use "detectors" or "neurons" in names
Examples of GOOD categories:
- Horror related
- French related
- Multilingual prepositions
- Medical terminology
- Multilingual unclear pattern
Categorize this neuron:
Sentence: {sentence}
Neuron: {neuron_description}
Existing categories: {existing_categories}

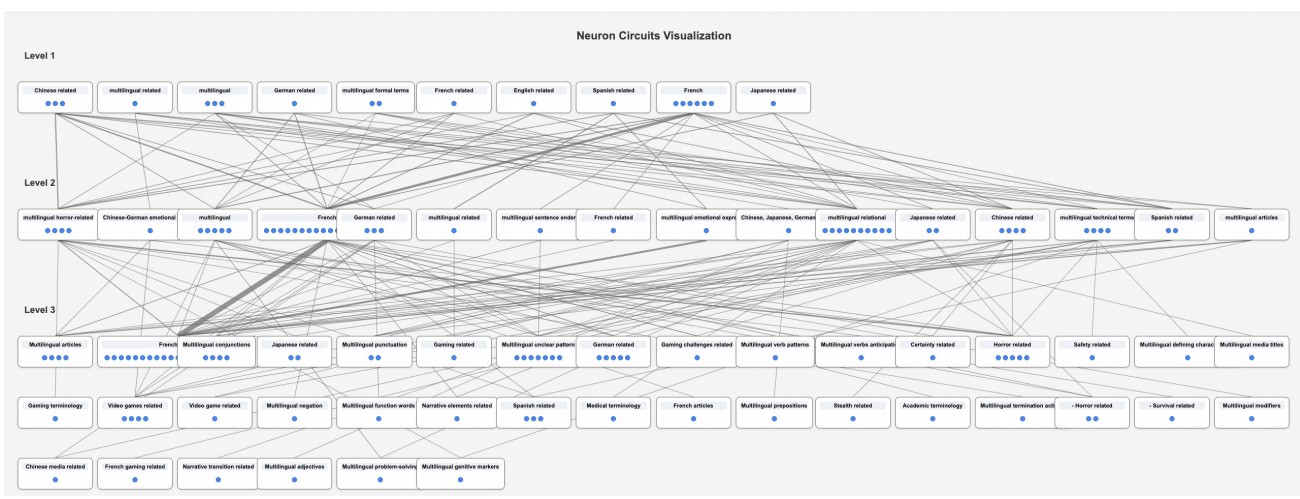

*Figure 11.* Complete circuit for normal French sample shown in Figure 2.

### A.4.3. CODE-SWITCHING EVALUATION

You are a language detector. Given a sentence and a specified language, your task is to determine whether the sentence contains any language other than the specified language. If it does, respond with YES. Containing a single word of another language should be considered YES. If it does not, respond with NO. Special terms such as keywords from specific languages, formulas, names of people, places, movies, numbers, or books in any language do not count as other languages. Consider the following examples:

Language: Chinese

Sentence: 最近，我在学习Python编程语言。它是一种强大的工具，适合处理各种数据分析任务，并且与NumPy和Pandas等库完美结合。因此，我打算在下个项目中使用Python来提高效率和生产力。

Response: NO

Language: English

Sentence: During my visit to Tokyo, I could not resist the aroma of新鲜制作的寿司。The flavors were so authentic and delicious that they reminded me of my childhood trips to Japan.

Response: YES

Language: French

Sentence: L'auteur du livre "Les Misérables", Victor Hugo, est considéré comme l'un des plus grands écrivains de la littérature française. Il a voyagé dans plusieurs pays européens, y compris l'Angleterre et l'Espagne, ce qui a influencé son œuvre.

Response: NO

Language: English

Sentence: I just finished a 杯of coffee when I decided to take a day off from work.

Response: YES

Language:$\{language\}$

Sentence: $\{sentence\}$

Response:

### A.5. Detailed Circuits

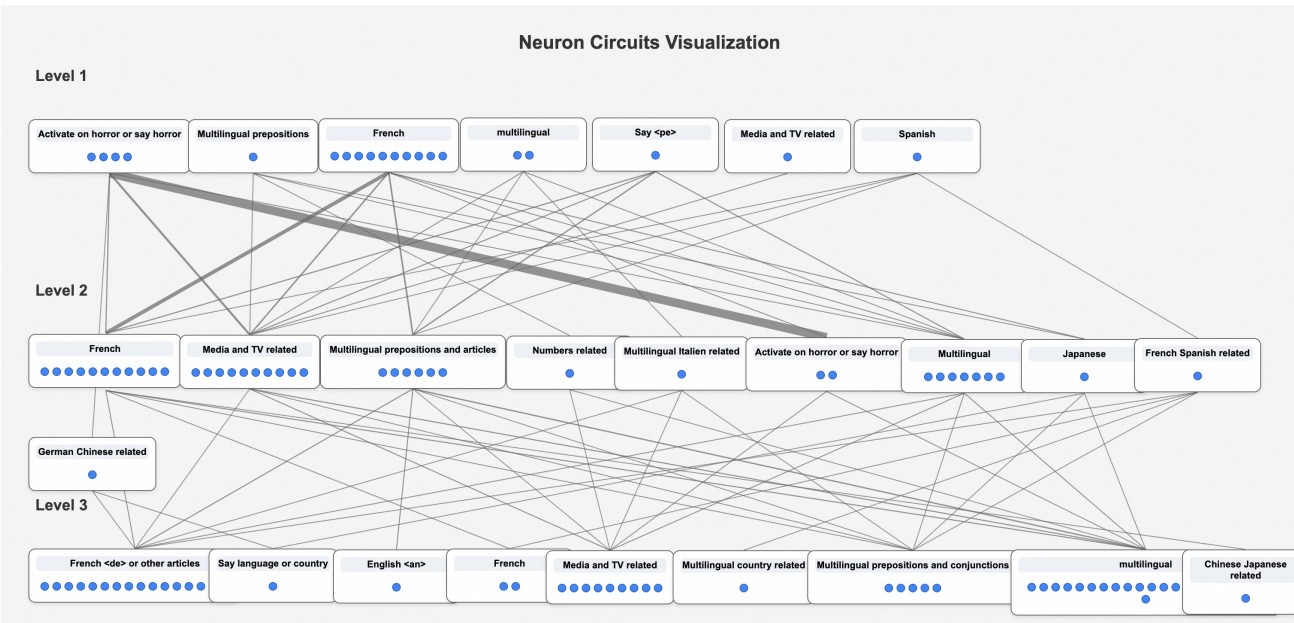

*Figure 12.* Complete circuit for code-switching on French-switched-to-Chinese sample shown in Figure 2.

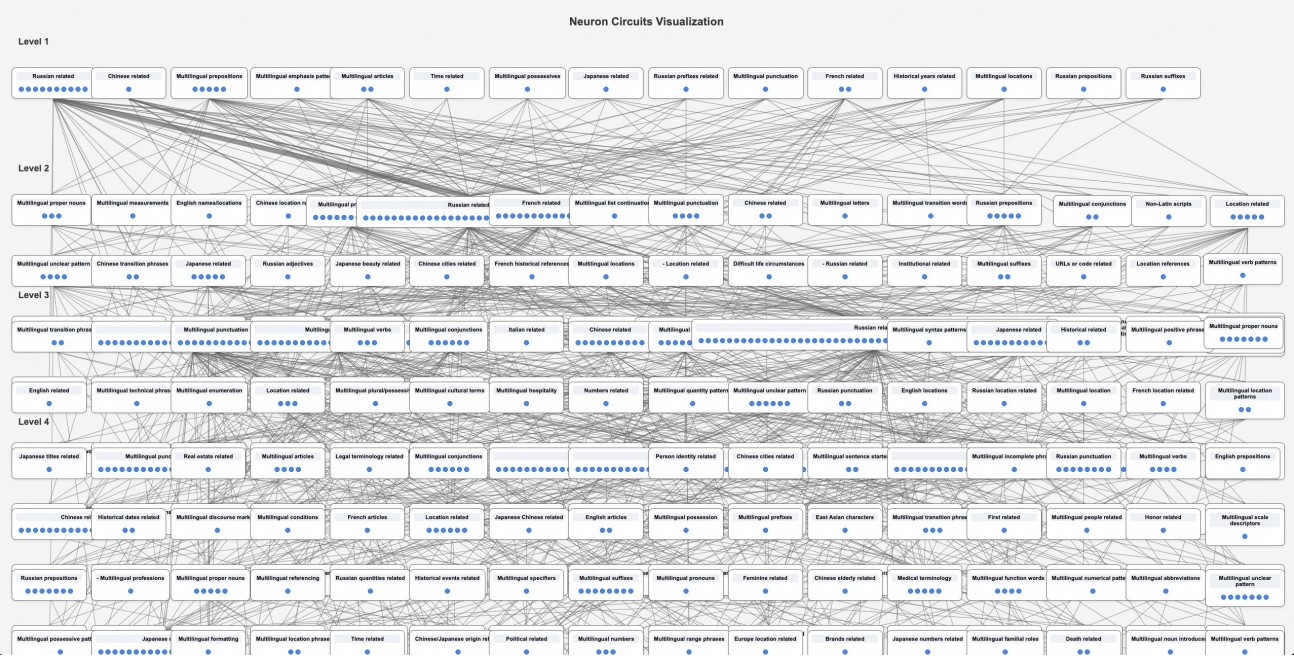

*Figure 13.* Complete circuit for Russian circuit.

