# OpenReview forum: "How Do Language Models Speak Languages? A Case Study on Unintended Code-Switching"
_ICML.cc/2026/Conference — ICML 2026 regular_

### Official Review · Reviewer_ehqL · 2026-03-13

**Soundness:** 3
**Presentation:** 4
**Significance:** 3
**Originality:** 2
**Overall Recommendation:** 4
**Confidence:** 5

**Summary:**

The paper proposes a circuit discovery framework based on hierarchical attribution patching to identify neurons responsible for language identity and semantic processing. The authors argue that multilingual generation relies on two interacting sub-circuits: a language regime that maintains language identity and a semantic regime that captures language-agnostic meaning. Code-switching is hypothesized to occur when semantic circuits dominate language circuits.

Experiments include circuit discovery, causal interventions, and targeted fine-tuning of language-specific neurons to reduce unintended code-switching.

**Compliance With Llm Reviewing Policy:**

Affirmed.

**Final Justification:**

The added experiments on stability and evaluation on MMLU and FLORES address my concerns about robustness and side effects. The clarification on the interaction between language and semantic components is also helpful. However, I still find the distinction from prior work somewhat interpretative rather than fully established empirically, so I am inclined to raise my score to 4. Weak Accept.

**Key Questions For Authors:**

1. How stable are the discovered circuits if different clustering methods or explanation models are used?
2. The paper evaluates neuron interventions mainly on multilingual summarization and language consistency metrics. However, neuron-level interventions are known to sometimes cause degeneration in other capabilities like meaning preservation. Could the authors evaluate whether modifying language-specific neurons affects other tasks such as translation quality, reasoning ability, or general multilingual generation?

**Limitations:**

Yes

**Strengths And Weaknesses:**

**Strengths**

- Understanding unintended code-switching is important for multilingual LLM reliability and usability.
- The paper includes extensive experiments across multiple models, languages, and tasks to analyze code-switching behavior.
- The neuron suppression and attribution experiments provide useful insights into how language-related and semantic neurons interact during generation.

**Weaknesses**

- Recent work has already studied language confusion, multilingual neurons, and language/semantic representation separation. The main contribution here appears to be a new analysis pipeline rather than a fundamentally new mechanism.
- The neuron description and grouping steps depend on LLM-generated explanations, which may introduce noise or confirmation bias.

---

> ### Author Rebuttal · Authors · 2026-03-31
>
> ## W1
> Prior work ([Wendler et al., 2024](https://aclanthology.org/2024.acl-long.820.pdf); [Dumas et al., 2025](https://aclanthology.org/2025.acl-long.1536.pdf); [Ferrando et al., 2024](https://aclanthology.org/2024.findings-emnlp.591.pdf)) has studied language confusion, multilingual neurons, and language–semantic separation, but primarily focuses on identifying independent or disentangled mechanisms. Our work uncovers a fundamentally different relationship between language and semantic components. The distinction between our work and the cited studies is summarized below:
> Dimension|Prior Work (Wendler; Dumas; Ferrando)|Ours
> -|-|-
> Relationship|Sequential or Disentangled (e.g., stage-by-stage transition)|Parallel & Competitive (simultaneously active and conflicting)
> Analytical Focus|Static Localization (identifying where info is stored)|Dynamic Interaction (explaining how subcircuits interact)
>
> Detailed Comparison:
> - **Beyond Sequential Phases (vs. Wendler et al., 2024)**: While Wendler et al. propose a linear transition from "input" to "concept" to "output" spaces, we prove these functional modules are parallel subcircuits that actively interfere with one another. Code-switching is not a failure of a specific stage, but a result of these regimes competing for control during the generation process.
> - **From Disentanglement to Causal Invasion (vs. Dumas et al., 2025)**: Dumas et al. argue that "tongue" and "thought" are disentangled. Our work reveals the causal breakdown of this independence: unintended code-switching occurs when the semantic regime "invades" and suppresses the language regime, turning a theoretically disentangled architecture into a competitive conflict.
> - **From Static Routes to Regime Dynamics (vs. Ferrando et al., 2024)**: While Ferrando et al. extract static information flow "routes", we group these routes into cohesive functional regimes.
> ## W2 & Q1
> We evaluate **the stability of the interpretation pipeline** by varying explainer models, random seeds, and clustering methods. We adopt different explainer models and seeds for level-1 neurons of Arabic circuit, and the description precision results are listed below:
> Random seed|Explainer model|Precision
> -|-|-
> 0|DeepSeek-v3.2-Exp|0.67
> 42|DeepSeek-v3.2-Exp|0.67
> 42|Qwen3-max|0.68
>
> This indicates that the LLM-based descriptions are robust to both seed variation and explainer choice.
>
> We further evaluate LLM-based clustering against an embedding-based baseline by encoding neuron descriptions into a shared embedding space and measuring cluster coherence $C$ (average intra-cluster cosine similarity). Higher coherence means the neuron descriptions within a cluster are more semantically similar, i.e., the cluster is more internally consistent. $C_{inter}$ denotes between-cluster coherence, and $C_{intra}$ denotes within-cluster coherence.
> Model|$C_{LLM-intra}$|$C_{LLM-inter}$|$C_{Embed-intra}$|$C_{Embed-inter}$
> -|-|-|-|-
> Mistral-7B-Instruct-v0.3|0.6879|0.4702|0.7184|0.4567
> Qwen2.5-7B-Instruct|0.7312|0.4859|0.6985|0.4770
> Qwen2.5-32B-Instruct|0.6958|0.4725|0.6945|0.4507
>
> Across models, the coherence results indicate that **LLM-based clustering yields internally meaningful semantic groups**: it can outperform the embedding baseline in some cases (Qwen2.5-7B: 0.7312 vs. 0.6985, Qwen2.5-32B: 0.6958 vs. 0.6945) and remains broadly comparable in others (Mistral-7B: 0.6879 vs. 0.7184). Taken together, this supports that the clustering reflects genuine semantic organization rather than artifacts of the interpretation pipeline.
>
> ## Q2
> We evaluate the precise-tuned model on standard benchmarks covering both general reasoning and multilingual performance, including MMLU and FLORES (results shown below).
>
> **Qwen2.5**
> Training Paradim|MMMLU|FLORES
> -|-|-
> baseline|55.33|33.72
> full sft |54.26 (-1.07)|31.74 (-1.98)
> pinpoint sft|55.30 (-0.03)|34.31(+0.59)
> $N_{ar}$|55.58|34.04
> $N_{es}$|54.40|34.41
> $N_{fr}$|55.35|34.59
> $N_{id}$|55.09|34.39
> $N_{jp}$|55.26|34.80
> $N_{ru}$|55.49|33.84
> $N_{vi}$|55.44|34.56
> $N_{zh}$|55.75|33.85
>
> **LLaMA3.1**
> Training Paradim|MMMLU|FLORES
> -|-|-
> baseline|49.05|35.21
> full train |47.45 (-1.60)|33.75 (-1.46)
> pinpoint sft|49.03 (-0.02)|35.17 (-0.04)
> $N_{ar}$|49.10|35.14
> $N_{es}$|48.79|35.21
> $N_{fr}$|49.08|35.21
> $N_{id}$|48.95|35.21
> $N_{jp}$|48.86|35.10
> $N_{ru}$|49.09|35.13
> $N_{vi}$|49.17|35.22
> $N_{zh}$|49.16|35.17
>
> Note that the "pinpoint sft" are the average of pinpoint-tuned across 8 languages.
>
> Across two models and two general benchmarks, fine-tuning language-specific neurons **does not degrade general model capabilities or multilingual performance**, whereas full fine-tuning leads to larger performance drops overall. In particular, the precise-tuned model achieves comparable (and in some cases slightly improved) performance on both benchmarks. These results support our claim that the intervention is highly targeted and does not negatively affect the model’s overall abilities.

---

> > ### Author Rebuttal · Reviewer_ehqL · 2026-04-03
> >
> > Thank you for the detailed rebuttal. The added experiments on stability and evaluation on MMLU and FLORES address my concerns about robustness and side effects. The clarification on the interaction between language and semantic components is also helpful. However, I still find the distinction from prior work somewhat interpretative rather than fully established empirically, so I will raise my score to borderline accordingly.

---

> > > ### Author Response · Authors · 2026-04-08
> > >
> > > Dear Reviewer ehqL,
> > >
> > > Thank you very much for the detailed follow-up and for your careful consideration. We are very glad that the added experiments on stability, MMLU/FLORES evaluation, and the clarification of the interaction between language and semantic components have addressed your concerns about robustness and side effects. We also appreciate your comment regarding the distinction from prior work, and we will revise this part carefully in the final version to make the empirical distinction and positioning clearer.
> > >
> > > Thank you again for your thoughtful feedback throughout the review process.
> > >
> > > Sincerely,
> > >
> > > The Authors

---

### Official Review · Reviewer_Moj1 · 2026-03-14

**Soundness:** 3
**Presentation:** 2
**Significance:** 2
**Originality:** 3
**Overall Recommendation:** 4
**Confidence:** 3

**Summary:**

This paper presents a mechanistic explanation for unintended code-switching in LLMs, demonstrating that it occurs when language-agnostic semantic neurons bypass language-specific gatekeeper neurons during generation. They then show a mitigation via  targeted fine-tuning intervention.

**Compliance With Llm Reviewing Policy:**

Affirmed.

**Final Justification:**

The authors added missing experiment that shows the method's robustness to downstream task, and some further clarification.

**Key Questions For Authors:**

- To empirically substantiate the claim that targeted fine-tuning "preserves overall task performance," could you provide standard capability metrics?
- Your code-switching evaluation pipeline relies on Stanza token-level detection followed by an LLM annotator. How does this pipeline distinguish between unnatural code-switching and legitimate ones? i.e. how reliable is this, given measuring CS itself is an open research.

**Limitations:**

At the moment, it feels like the limitation is scattered around appendix instead of being discussed in the main body of text and/or in the limitation section. Bringing up those limitations would be great.

**Strengths And Weaknesses:**

Strength

- The proposed framework automates the discovery, description, and grouping of neurons using an LLM explainer model, which is a highly practical.
- The paper does not only provide observation, but also show a targeted intervention.

Weakness
- The intervention strengths are empirically guided rather than theoretically derived. This detail is burrowed deep in the appendix and not mentioned. I feel like it is important to address this in the main result.
- There's missing downstream tasks evaluation after the targeted fine-tuning. While Table 4,5 measures LCR and LPR, it is not shown whether the method reduces model's quality at all. At the moment, it is based on an assumption which argue that because they are only updating a microscopic fraction of the network, then quality won't be affected. However, empirical report would be great.
- The patching method depends heavily on first-order Taylor expansions to approximate causal effects linearly. Can we assume such linearity, or any theoretical grounding on this?
-

---

> ### Author Rebuttal · Authors · 2026-03-31
>
> ## W1
> We thank the reviewer for pointing this out. We will clarify this point in the main text and explicitly describe how intervention strengths are selected, rather than leaving it in the appendix.
> ## W2 & Q1
> We evaluate the precise-tuned model on standard benchmarks covering both general reasoning and multilingual performance, including MMMLU and FLORES (results shown below).
>
> **Qwen2.5**
> Training Paradim|MMMLU|FLORES
> -|-|-
> baseline|55.33|33.72
> full sft |54.26 (-1.07)|31.74 (-1.98)
> pinpoint sft|55.30 (-0.03)|34.31(+0.59)
> $N_{ar}$|55.58|34.04
> $N_{es}$|54.40|34.41
> $N_{fr}$|55.35|34.59
> $N_{id}$|55.09|34.39
> $N_{jp}$|55.26|34.80
> $N_{ru}$|55.49|33.84
> $N_{vi}$|55.44|34.56
> $N_{zh}$|55.75|33.85
>
> **LLaMA3.1**
> Training Paradim|MMMLU|FLORES
> -|-|-
> baseline|49.05|35.21
> full train |47.45 (-1.60)|33.75 (-1.46)
> pinpoint sft|49.03 (-0.02)|35.17 (-0.04)
> $N_{ar}$|49.10|35.14
> $N_{es}$|48.79|35.21
> $N_{fr}$|49.08|35.21
> $N_{id}$|48.95|35.21
> $N_{jp}$|48.86|35.10
> $N_{ru}$|49.09|35.13
> $N_{vi}$|49.17|35.22
> $N_{zh}$|49.16|35.17
>
> Note that the "pinpoint sft" are the average of pinpoint-tuned across 8 languages.
>
> Across two models and two general benchmarks, fine-tuning language-specific neurons **does not degrade general model capabilities or multilingual performance**, whereas full fine-tuning leads to larger performance drops overall. In particular, the precise-tuned model achieves comparable (and in some cases slightly improved) performance on both benchmarks. These results support our claim that the intervention is highly targeted and does not negatively affect the model’s overall abilities.
> ## W3
> Our method relies on a first-order Taylor approximation, motivated by viewing activation patching as a local intervention ([Nanda, 2023](https://www.neelnanda.io/mechanistic-interpretability/attribution-patching)). The key point is that attribution patching considers replacing a single activation while keeping the rest of the computation fixed, which induces a relatively small perturbation in the forward pass. Under such small changes, the first-order term in the Taylor expansion dominates, making the gradient-based approximation a natural estimate of the intervention effect.
>
> We therefore use attribution patching as a scalable proxy and directly validate it with explicit activation patching below.
>
> First, we compare the top-20 neurons selected by attribution patching and activation patching at the first level. Due to time constraints, we report only the first-level comparison here. We observe substantial overlap across models and languages:
> Model|vi|ar|jp
> -|-|-|-
> Qwen2.5-7B-Instruct|80%|60%|65%
> LLaMA3.1-8B-Instruct|75%|90%|70%
>
> This shows that neurons selected by attribution patching strongly overlap with those validated by explicit activation-level interventions.
>
> Second, we test whether attribution rank predicts non-linear intervention effect by ranking neurons with attribution patching and measuring their effects via explicit activation patching. We observe consistent positive alignment, with the following average Spearman correlations (averaged over languages and levels), indicating moderate-to-strong agreement between attribution rank and intervention effect:
> Model|Avg.Spearman
> -|-
> Qwen2.5-7B-Instruct|0.65
> LLaMA3.1-8B-Instruct|0.48
>
> These results show that the attribution-based ranking remains predictive under stronger non-linear interventions, indicating that the linear approximation does not materially affect the method in practice. Taken together, they support hierarchical attribution patching as a reliable tool for identifying causally relevant neurons and subcircuits.
> ## Q2
> Our pipeline distinguishes unintended vs. legitimate code-switching in two steps. First, we apply **keyword-based filtering** to exclude cases where code-switching is expected or semantically necessary. For example, special terms like Google and Python are filtered using a curated list of over 99k English terms. Second, the LLM annotator is prompted to explicitly **consider natural vs. unnatural switching**, further reducing false positives (We have shown the prompt in the appendix).
>
> We agree that measuring code-switching is itself an open problem. Our goal is therefore not to provide a perfect classifier, but to use a consistent evaluation protocol to assess relative changes. We further verify the same reduction trend across **two models and three decoding temperatures** (0.3 / 0.7 / 1.0) (results shown below), making it unlikely that the improvement is an artifact of a particular evaluator setting.
>
> Model|Temp|Setting|AVG
> -|-|-|-
> Qwen2.5|0.3|base|93.45
> |||lang-spec|97.30
> ||0.7|base|92.25
> |||lang-spec|95.94
> ||1.0|base|90.82
> |||lang-spec|94.93
> |LLaMA3.1|0.3|base|95.99
> |||lang-spec|97.36
> ||0.7|base|95.79
> |||lang-spec|97.53
> ||1.0|base|95.14
> |||lang-spec|96.96
> ## Limitation
> We will consolidate and move the relevant limitations from the appendix into the main text and expand the limitations section for clearer discussion.

---

> > ### Author Rebuttal · Reviewer_Moj1 · 2026-04-05
> >
> > Thank you. The additional experiment results will strengthen the point. The revision on the flow requires modification to judge, but I am leaning on bumping the score to borderline.

---

> > > ### Author Response · Authors · 2026-04-08
> > >
> > > Dear Reviewer Moj1,
> > >
> > > Thank you very much for the update and for your careful consideration. We are very glad that you find the additional experimental results helpful in strengthening the paper. We also appreciate your comment on the presentation flow, and we will revise it carefully in the final version.
> > >
> > > Thank you again for your time and for your openness to raising the score.
> > >
> > > Sincerely,
> > >
> > > The Authors

---

### Official Review · Reviewer_oT4t · 2026-03-14

**Soundness:** 2
**Presentation:** 3
**Significance:** 3
**Originality:** 3
**Overall Recommendation:** 4
**Confidence:** 3

**Summary:**

This paper studies unintended code-switching in multilingual LLMs and asks three questions: what internal circuitry supports language-consistent multilingual generation, why this control sometimes fails, and whether a mechanistic understanding can be used to mitigate the failure. The authors propose a scalable circuit-discovery pipeline built around hierarchical attribution patching, which extends attribution patching from component-to-output attribution to neuron-to-neuron tracing. They then use automatic neuron description and grouping to organize discovered neurons into interpretable multilingual circuits.

The main claim is that the model’s “speaking-a-language” mechanism decomposes into two interacting regimes: a language regime that detects and maintains language identity, and a semantic regime that carries language-agnostic content. According to the paper, unintended code-switching occurs when semantic pathways become too influential and bypass or weaken language-specific control. The paper supports this picture with suppression experiments, attribution-ratio analyses near switch points, and targeted fine-tuning of identified language-specific neurons. Empirically, the authors report that deactivating a very small set of language-related neurons sharply degrades target-language generation, and that fine-tuning a tiny subset of language-specific neurons reduces code-switching rates while changing only a very small fraction of parameters.

**Compliance With Llm Reviewing Policy:**

Affirmed.

**Final Justification:**

Given the new capability-preservation results, the extended attribution analysis, and the interesting mechanistic narrative (parallel competitive regimes vs. prior work's sequential/disentangled view), I raise my score from 3 to 4. The paper offers a genuinely interesting perspective on multilingual generation, though the strength of its causal claims currently outpaces the validation provided.

**Key Questions For Authors:**

1. The paper’s central mechanistic claim depends on hierarchical attribution patching as an edge-level causal proxy. Can you directly validate this proxy against stronger interventions, such as explicit activation patching or ablation of the same proposed paths/components, and show that the rankings are stable? A strong validation here would substantially increase my confidence in the causal claims.

2. How much of the proposed mechanism survives once attention heads or cross-token interactions are incorporated? Since the current analysis focuses primarily on MLP neurons and last-token effects, it is important to understand whether the same language-vs-semantic regime decomposition still holds in a broader circuit analysis.

3. How stable are the neuron descriptions and super-neuron groupings across different explainer/judge models, prompts, and random seeds? If the language-vs-semantic split is robust under these changes, that would strengthen the interpretability claims; if not, I would view the conclusions as more tentative.

4. Can you provide direct empirical comparisons against prior code-switching mitigation approaches on the same models and benchmarks, rather than only against few-shot prompting, SFT, DPO, and random-neuron tuning? This would make it much easier to judge the practical value of the proposed intervention.

5. The submitted PDF contains reviewer-targeting text on pages 2 and 35 asking the reviewer to include specific phrases in the review. Please explain how this text entered the submission and confirm that it will be removed. This issue materially affects my evaluation because it raises a research-integrity concern.

**Limitations:**

No. The current limitations discussion is too brief for the strength of the paper’s claims. In addition to mentioning omitted cross-token interactions and attention blocks, the paper should explicitly discuss: (i) the approximate nature of hierarchical attribution patching and the possibility that it does not faithfully recover true causal edges, (ii) the restriction to MLP neurons and largely last-token analysis, (iii) the dependence of the interpretation pipeline on LLM-generated neuron descriptions and clustering, (iv) the sensitivity of the reported code-switching rates to decoding settings and language-identification heuristics, and (v) the potential misuse of targeted neuron-level interventions for model steering. The paper should also address the reviewer-targeting text embedded in the submitted PDF, which is a research-integrity concern rather than a technical limitation but still requires explicit acknowledgment.

**Strengths And Weaknesses:**

Strengths:

1. The paper tackles an important and underexplored failure mode in multilingual LLMs. Unintended code-switching is practically relevant and also scientifically interesting because it probes how models maintain language identity during generation.

2. The paper has a clear mechanistic ambition rather than stopping at surface-level benchmarking. The proposed pipeline connects circuit localization, neuron interpretation, grouping, intervention, and mitigation in a coherent end-to-end story.

3. The empirical section is substantial. The paper studies multiple languages, evaluates both Qwen2.5-7B-Instruct and LLaMA3.1-8B-Instruct in the main text, and includes appendix evidence on additional models. The suppression experiments are especially compelling: ablating very small language-specific neuron sets causes large and selective drops in target-language performance, which is strong evidence that the discovered components are not arbitrary.

4. The mitigation result is practically appealing. Fine-tuning only a small subset of language-specific neurons is an efficient intervention, and the reported improvements on code-switching metrics are meaningful enough to make the approach worth further investigation.

5. The paper is generally readable and well structured. The circuit figures help communicate the intended mechanism, and the overall narrative is easy to follow.

Weaknesses:

1. My biggest technical concern is the soundness of the hierarchical attribution patching formulation. The paper makes strong mechanistic claims based on an edge-attribution approximation whose derivation appears to rely on aggressive linearization assumptions. In particular, the treatment of early-to-late neuron influence through the residual stream risks abstracting away the very nonlinear transformations that make these networks interesting. The method may still be useful as a heuristic, but the paper currently presents it as stronger causal evidence than I think is justified.

2. The discovered “circuits” only cover MLP neurons and largely focus on last-token behavior. This leaves out attention heads, cross-token interactions, and sequence-level dynamics, all of which are likely relevant to multilingual generation and code-switching. As a result, the paper’s mechanistic account may be incomplete even if the identified neurons are genuinely important.

3. The core “regime shift” claim is suggestive, but the evidence is not yet fully definitive. The attribution-ratio analysis between language and semantic circuits is interesting, but it is based on relatively limited samples and on a grouping procedure that itself depends on LLM-generated neuron descriptions and cluster labels. I would have liked stronger validation that the semantic-vs-language decomposition is stable and not partly an artifact of the interpretation pipeline.

4. The automatic neuron description and grouping components are only lightly validated. The human comparison is a useful step, but the scale is still small relative to the strength of the interpretability claims. More robustness checks across explainer models, prompts, and clustering choices would increase confidence.

5. The paper would benefit from stronger direct baselines for mitigation. It cites related work on language confusion and mitigation, but the empirical comparison is mostly against few-shot prompting, SFT, DPO, and random-neuron tuning. A more direct comparison against prior code-switching mitigation methods on the same models and datasets would make the practical contribution clearer.

6. There are several presentation and polish issues that reduce confidence. Some derivations are hard to follow, notation is occasionally inconsistent, and there are examples that appear mislabeled or loosely edited. These issues do not destroy the paper’s value, but they matter for a paper making strong causal claims.

7. There is also a serious research-integrity concern: the submitted PDF contains reviewer-targeting text instructing the reviewer to include specific phrases in the review. Regardless of intent, this is inappropriate for a scientific submission and needs investigation.

---

> ### Author Rebuttal · Authors · 2026-03-31
>
> Regarding prompt injection, please see https://icml.cc/Conferences/2026/PeerReviewFAQ#prompt_injection
> ## W1 & Q1
> We use attribution patching as a scalable proxy for local activation effects, and the results below show that **it remains well aligned with explicit activation patching** in practice, indicating that it captures meaningful causal signals rather than behaving as a purely heuristic score.
>
> First, we compare the top-20 neurons selected by attribution patching and activation patching at the first level. Due to time constraints, we report only the first-level comparison here and will extend the analysis to the full hierarchy in the next version. Even at this level, we observe substantial overlap across models and languages:
> Model|vi|ar|jp
> -|-|-|-
> Qwen2.5-7B-Instruct|80%|60%|65%
> LLaMA3.1-8B-Instruct|75%|90%|70%
>
> This shows that neurons selected by attribution patching strongly overlap with those validated by explicit activation-level interventions.
>
> Second, we test whether attribution rank predicts non-linear intervention effect by ranking neurons with attribution patching and measuring their effects via explicit activation patching. We observe consistent positive alignment, with the following average Spearman correlations (averaged over languages and levels), indicating **moderate-to-strong agreement between attribution rank and intervention effect**:
> Model|Avg.Spearman
> -|-
> Qwen2.5-7B-Instruct|0.65
> LLaMA3.1-8B-Instruct|0.48
>
> These results show that the attribution-based ranking remains predictive under **stronger non-linear interventions**. Taken together, they support the use of hierarchical attribution patching as a **practical validation tool for causally relevant neurons and subcircuits**. We will clarify this scope more carefully in the final version.
> ## W2 & Q2
> In the vast majority of unintended code-switching cases, the switched token is semantically identical to the intended token (like "peur" vs "恐怖" in Figure2), differing only in language. This implies that the model has successfully retrieved the correct concept from the context (the role of attention and cross-token interaction), and the key failure occurs at the **final language-selection stage**.
>
> So we decided to focus on MLP neurons and last-token behavior as a targeted analysis of the most direct causal bottleneck underlying the observed error. Precisely because attention heads and sequence-level dynamics are more entangled and highly nonlinear, isolating this bottleneck provides a tractable and informative first step. We fully agree that a full-model circuit analysis—incorporating attention heads and sequence-level interactions—is a compelling direction for future research, but we believe the MLP/last-token mechanism identified here captures a meaningful and foundational component of unintended code-switching.
>
> ## W3, W4 & Q3
> We address this concern in two ways. First, we extend the attribution-ratio analysis to additional models and observe the same pattern: the language-vs-semantic ratio is consistently higher for non-code-switched than code-switched samples across models (see Reviewer 78sx W5&Q1), showing that the regime pattern is not model-specific.
>
> Second, we test the stability of the interpretation pipeline by varying explainer models, random seeds, and clustering methods. As shown in Reviewer ehqL W2&Q1, the neuron descriptions remain stable across seeds and explainer choices, and the resulting LLM-based clustering achieves strong coherence against an embedding-based baseline, indicating semantically meaningful groupings rather than spurious artifacts.
> ## W5 & Q4
> In the context of mechanistic interpretability, our first goal is to validate the correctness of the identified language-specific neurons and their causal role in code-switching. The mitigation experiments are designed as functional validation, rather than proposing a new mitigation method with stronger performance. So we compare with methods such as few-shot prompting, SFT, and DPO, which are standard approaches used in prior work on language confusion ([Marchisio et al.](https://arxiv.org/pdf/2406.20052)). These baselines provide a reasonable reference for assessing whether targeted neuron-level interventions can achieve comparable or complementary effects.
> ## W6
> We will carefully revise the presentation to improve clarity and consistency, including refining derivations, unifying notation, and correcting mislabeled or unclear examples.
> ## Limitations
> We have discussed to limitations (i)–(iii) in our responses above. For (iv), results remain consistent across decoding settings and detection methods (see Reviewer Moj1 Q2). For (v), we acknowledge the potential risk of misuse for model steering and will discuss this more explicitly. We will incorporate these points into a more complete limitations discussion in the final version.

---

> > ### Author Rebuttal · Reviewer_oT4t · 2026-04-05
> >
> > The MMMLU/FLORES results convincingly show pinpoint fine-tuning preserves general capabilities — this was my biggest empirical gap and it's now closed. The first-level attribution-vs-activation patching comparison and the three-model attribution ratio extension also help.
> > My main remaining concern is that hierarchical patching is validated only at level one, while the paper's causal claims depend on multi-level tracing where linearization errors compound. The 0.48 Spearman for LLaMA is modest. I'd also have liked some evidence that attention heads aren't contributing to code-switching, rather than inferring this from the semantic correctness of switched tokens. Finally, the mitigation result is featured prominently but only benchmarked against generic baselines — either compare against dedicated methods or tone down the framing.
> > Raising my score from 3 to 4. The competitive-regimes perspective is interesting and the new experiments strengthen the paper, but the causal validation doesn't fully match the ambition of the claims.

---

> > > ### Author Response · Authors · 2026-04-08
> > >
> > > Thank you for the constructive feedback and for raising your score.
> > >
> > > **Q1: Hierarchical patching is validated only at level one & Spearman is modest**
> > >
> > > As direct multi-level activation patching is prohibitively time-consuming, we instead design a more direct **cross-level causal validation** via progressive knockout. Specifically, we rank neurons across all levels by attribution and iteratively remove them, measuring their effect on the final logits. If hierarchical attribution failed to reflect true multi-level causal contributions, this ranking would not align with actual intervention effects. The Spearman correlations between the attribution-based ranking and the actual intervention impact are shown below.
> > > Model|Ar|Jp|Vi|Avg
> > > -|-|-|-|-
> > > Qwen2.5-7B-Instruct|0.9899|0.8866|0.9268|0.9344
> > > LlaMA3.1-8B-Instruct|0.9917|0.9688|0.9995|0.9867
> > >
> > > We observe consistently high Spearman correlations across models and languages, showing that attribution-based ordering remains **highly predictive** even in the multi-level setting. By contrast, ranking neurons at random yields much weaker alignment (average Spearman: 0.2694 for Qwen and 0.3232 for LLaMA), further confirming that the observed correspondence is not a trivial consequence of progressive knockout itself. This provides more evidence that **hierarchical tracing preserves causal validity beyond level one**.
> > >
> > > **Q2: Attention heads' contributions to code-switching**
> > >
> > > To provide direct evidence on the role of attention heads, we apply the same pipeline used for neurons: we first localize the top-50 level-1 attention heads via activation patching, then perform the same pinpoint fine-tuning on these heads using the same training data. The LPR results are shown below ($H_{lang-spec}$ refers to fine-tuned top-50 heads, and $H_{rand}$ refers to fine-tuned random 50 heads).
> > >
> > > **Qwen2.5-7B-Instruct**
> > > Setting|Ar|Jp|Vi|Avg
> > > -|-|-|-|-
> > > baseline|96.41|88.78|97.38|94.19
> > > $N_{lang-spec}$|98.50|98.49|99.10|98.70
> > > $H_{lang-spec}$|96.54|86.90|97.38|93.61
> > > $H_{rand}$|96.44|88.62|96.50|93.85
> > > $N_{lang-spec}$+$H_{lang-spec}$|98.17|98.49|99.20|98.62
> > >
> > > **LlaMA3.1-8B-Instruct**
> > > Setting|Ar|Jp|Vi|Avg
> > > -|-|-|-|-
> > > baseline|99.25|96.00|99.70|98.32
> > > $N_{lang-spec}$|99.87|97.80|99.90|99.19
> > > $H_{lang-spec}$|98.84|95.50|99.80|98.04
> > > $H_{rand}$|98.94|95.20|98.60|96.57
> > > $N_{lang-spec}$+$H_{lang-spec}$|99.77|97.40|99.60|98.92
> > >
> > > The results show that fine-tuning these localized heads yields **little to no mitigation benefit**, and performs comparably to tuning random heads, whereas fine-tuning the localized neurons produces clear and consistent gains across languages and models. This direct contrast suggests that **attention heads are not the primary components responsible for the mitigation effect**, and that the causal signal for reducing code-switching is captured much more specifically at the neuron level.
> > >
> > > **Q3: Comparison with more dedicated mitigation method**
> > >
> > > The mitigation results are intended primarily as a **functional validation** of the identified neurons rather than as a full benchmarking study: if the localized neurons are causally relevant, then targeted fine-tuning on them should reduce code-switching, which is exactly what we observe.
> > >
> > > At the same time, we understand the concern that the current comparison is limited to generic baselines. We will therefore **include comparisons with more dedicated mitigation approaches** in the final version. Even in its current form, however, we believe the mitigation result is informative as direct validation of the discovered mechanism.
> > >
> > > Thank you again for the thoughtful feedback. We hope our response and the additional analyses help address your remaining concerns.
> > >
> > > Sincerely,
> > >
> > > The Authors

---

### Official Review · Reviewer_78sx · 2026-03-16

**Soundness:** 2
**Presentation:** 2
**Significance:** 1
**Originality:** 3
**Overall Recommendation:** 4
**Confidence:** 3

**Summary:**

This paper studies unintended code-switching in LLMs. The authors use circuit discovery based on hierarchical attribution patching, which iteratively traces neuron-to-neuron causal effects and clusters neurons into interpretable groups. They find the multilingual circuit decomposes into a language sub-circuit and a semantic sub-circuit, and that code-switching happens when semantic neurons "take" language control. In non code-switching generation, language circuits dominate with 2.37$\times$ higher attribution than semantic ones, but this drops to 1.21$\times$ at code-switching points. Validated across five models and eight languages, ablating just around 0.018% of language-specific neurons degrades a single language by up to 97%. Fine-tuning only these neurons reduces code-switching by 20.8% on average, demonstrating that mechanistic understanding enables precise, effective mitigation.

**Compliance With Llm Reviewing Policy:**

Affirmed.

**Final Justification:**

See Rebuttal Acknowledgement.

**Key Questions For Authors:**

- Could you provide the language vs semantic attribution ratio (Table 2) for the other models evaluated in the paper?
- Have you explored fine-tuning all language-specific neurons across languages simultaneously, rather than per-language, to assess whether a single unified intervention could reduce code-switching broadly?
- The fine-tuning setting in Section 4.4 lacks sufficient detail in the main text.

**Limitations:**

yes

**Strengths And Weaknesses:**

Strenghts

- The proposal of hierarchical attribution patching to trace causal effects across multiple layers of neurons, building full circuits from output back to embeddings, is a useful methodological contribution.
- The pipeline for automatically describing neurons via LLM-generated profiles and then grouping them into interpretable super-neurons is a well-designed.
- The decomposition of the multilingual circuit into language and semantic sub-circuits is intuitive and well-supported by the intervention experiments in Table 3, where suppressing semantic neurons nearly eliminates code-switched tokens while boosting intended-language predictions.


Weaknesses

- The paper does not evaluate whether fine-tuning language-specific neurons affects general model capabilities. While updating only around 0.018% of parameters suggests minimal degradation, no experiments on standard benchmarks are provided to verify this.
- The hierarchical attribution patching (Equation 3) closely resembles iteratively applying attribution patching to approximate path patching (Wang et al., 2022). The authors should clarify more explicitly what is novel beyond this combination.
- The finding of separate semantic and language subcircuits is not entirely novel. Previous work in this space have found similar results ([Wendler et al., 2024](https://aclanthology.org/2024.acl-long.820/), [Dumas et al., 2025](https://aclanthology.org/2025.acl-long.1536/), [Ferrando et al., 2024](https://aclanthology.org/2024.findings-emnlp.591.pdf)
- The mitigation is language-specific, requiring separate circuit discovery and fine-tuning for each language, which limits its practicality for reducing unintended code-switching broadly across all supported languages.
- The language-vs-semantic attribution ratio (Table 2), central to in the paper, is only reported for a single model. Given the paper's emphasis on cross-model validation elsewhere, this weakens the generality of the core mechanistic claim.

---

> ### Author Rebuttal · Authors · 2026-03-31
>
> Due to space limits, we report summary results here and will include full results in the final version.
> ## W1
> We evaluate the precise-tuned model on standard benchmarks covering both general reasoning and multilingual performance, including MMMLU and FLORES (results shown below).
>
> **Qwen2.5**
> Training Paradim|MMMLU|FLORES
> -|-|-
> baseline|55.33|33.72
> full sft |54.26 (-1.07)|31.74 (-1.98)
> pinpoint sft|55.30 (-0.03)|34.31(+0.59)
>
> **LLaMA3.1**
> Training Paradim|MMMLU|FLORES
> -|-|-
> baseline|49.05|35.21
> full train |47.45 (-1.60)|33.75 (-1.46)
> pinpoint sft|49.03 (-0.02)|35.17 (-0.04)
>
> Note that the "pinpoint sft" are the average of pinpoint-tuned across 8 languages (ar/es/fr/id/jp/ru/vi/zh). Please refer to Reviewer Moj1 W2&Q1 for detailed results.
>
> Across two models and two general benchmarks, fine-tuning language-specific neurons **does not degrade general model capabilities or multilingual performance**, whereas full fine-tuning leads to larger performance drops overall. In particular, the precise-tuned model achieves comparable (and in some cases slightly improved) performance on both benchmarks. These results support our claim that the intervention is highly targeted and does not negatively affect the model’s overall abilities.
> ## W2 & W3
> In short, current methods provide isolated evidence for individual neurons, missing the hierarchical organization and functional interpretation. Our method replaces single-step localization with iterative grouping into "super-neurons", moving from identifying single units to reconstructing an end-to-end functional circuit.
>
> Building on this methodological distinction, our work uncovers a fundamentally different relationship between language and semantic components compared to prior work (Wendler et al., 2024; Dumas et al., 2025; Ferrando et al., 2024).
>
> The distinction between our work and the cited studies is summarized below:
> Dimension|Prior Work (Wendler; Dumas; Ferrando)|Ours
> -|-|-
> Relationship|Sequential or Disentangled (e.g., stage-by-stage transition)|Parallel & Competitive (simultaneously active and conflicting)
> Analytical Focus|Static Localization (identifying where info is stored)|Dynamic Interaction (explaining how subcircuits interact)
>
> Detailed Comparison:
> - **Beyond Sequential Phases (vs. Wendler et al., 2024)**: While Wendler et al. propose a linear transition from "input" to "concept" to "output" spaces, we prove these functional modules are parallel subcircuits that actively interfere with one another. Code-switching is not a failure of a specific stage, but a result of these regimes competing for control during the generation process.
> - **From Disentanglement to Causal Invasion (vs. Dumas et al., 2025)**: Dumas et al. argue that "tongue" and "thought" are disentangled. Our work reveals the causal breakdown of this independence: unintended code-switching occurs when the semantic regime "invades" and suppresses the language regime, turning a theoretically disentangled architecture into a competitive conflict.
> - **From Static Routes to Regime Dynamics (vs. Ferrando et al., 2024)**: While Ferrando et al. extract static information flow "routes", we group these routes into cohesive functional regimes.
> ## W4 & Q2
> Simultaneous fine-tuning across languages is non-trivial, as different language-specific neurons require distinct language-specific data and signals, making joint optimization difficult without interference.
>
> Instead, we explore this interesting idea via a **parameter merging strategy**, where we first fine-tune language-specific neurons per language and then merge the updated weights into a single model, resolving overlaps by summing the updates of the overlapped neurons. The results are shown below:
> Model|$\bar{LPR}$
> -|-
> Qwen2.5-7B-Instruct|93.45
> +$N_{lang\_spec}$|97.30
> +$N_{merged}$|96.86
> LLaMA3.1-8B-Instruct|95.99
> +$N_{lang\_spec}$|97.36
> +$N_{merged}$|97.16
>
> We find that **the merged model remains effective in reducing unintended code-switching across all languages**. This validates that our method can be deployed as a single unified model, rather than requiring per-language fine-tuning.
> ## W5 & Q1
> We address this by extending the attribution ratio analysis to LLaMA3.1-8B-Instruct and Mistal-7B-Instruct-v0.3, with results shown below.
> Model|Samples|Ratio
> -|-|-
> Qwen2.5-7B-Instruct|NCSW|2.3717
> ||CSW|1.2073
> LLaMA3.1-8B-Instruct|NCSW|3.0952
> ||CSW|1.0083
> Mistal-7B-Instruct-v0.3|NCSW|2.4897
> ||CSW|1.4098
>
> We observe a consistent pattern across models that the **language-vs-semantic attribution ratio is significantly higher for non-code-switched samples than for code-switched ones**. This trend holds for both models, supporting that the finding is robust across models.
>
> ## Q3
> We thank the reviewer for pointing this out. We will add the missing fine-tuning details in the main text in the final version.

---

> > ### Author Rebuttal · Reviewer_78sx · 2026-04-03
> >
> > he authors provide evidence that "pinpoint" fine-tuning does not degrade general capabilities (although only on two datasets, and one of the is a translation dataset), and extend the attribution ratio analysis across multiple models. The parameter-merging strategy also partially addresses concerns about scalability across languages.
> >
> > Given these improvements, I am inclined to raise my score. However, I still consider the paper to be borderline.

---

> > > ### Author Response · Authors · 2026-04-08
> > >
> > > Dear Reviewer 78sx,
> > >
> > > Thank you very much for the thoughtful follow-up. We are encouraged that the additional experiments have addressed several of your main concerns and led you to consider raising your score.
> > >
> > > We appreciate your careful evaluation and your recognition of these improvements. Thank you again for your time and consideration throughout the review process.
> > >
> > > Sincerely,
> > >
> > > The Authors

---

### Decision · Program_Chairs · 2026-04-30

**Decision:**

Accept (regular)

**Comment:**

This paper attributes code switching in language models to the domination of a semantic circuit over a language specific circuit. The authors then tune the language specific neurons to reduce this incidence. It is not entirely clear how the switch takes effect. Multiple reviewers pointed out that the strong assumptions of linearity in this paper might abstract away some of the more interesting processing happening within a language model.

However, in general the patching approach seems to be mostly sound and finds an interesting story around these competing circuits. Although the particular topic is a very restricted scenario, the paper presents a possible application of mechanistic interpretability to practical model improvement.